# Intern-GS: Vision Model Guided Sparse-View 3D Gaussian Splatting

## Abstract

Sparse-view scene reconstruction often faces significant challenges due to the constraints imposed by limited observational data. These limitations result in incomplete information, leading to suboptimal reconstructions using existing methodologies. To address this, we present **Intern-GS**, a novel approach that effectively leverages rich prior knowledge from vision foundation models to enhance the process of sparse-view Gaussian Splatting, thereby enabling high-quality scene reconstruction. Specifically, Intern-GS utilizes vision foundation models to guide both the initialization and the optimization process of 3D Gaussian splatting, effectively addressing the limitations of sparse inputs. In the initialization process, our method employs DUSt3R first to generate a dense gaussian point cloud. This approach significantly alleviates the limitations encountered by traditional structure-from-motion (SfM) methods, which often struggle under sparse-view constraints. However, directly using DUST3R tends to introduce unnecessary redundancy. To mitigate this, we propose a redundancy-free strategy that leverages confidence scores to remove overlapping regions across frames. During the optimization process, we propose a hybrid regularization strategy that jointly constrains both observed and unobserved views in terms of color and geometry, guiding 3DGS optimization toward more accurate reconstructions. Extensive experiments demonstrate that Intern-GS achieves state-of-the-art rendering quality across diverse datasets, including both forward-facing and large-scale scenes, such as LLFF, DTU, and Tanks and Temples.

## 1 Introduction

Novel View Synthesis (NVS) Zhou et al. (2016); Avidan & Shashua (1997) focuses on generating images from unseen perspectives using a series of images captured from specific seen viewpoints. This technology has significant potential in applications such as virtual reality (VR), film production, urban planning, autonomous driving, and game design, establishing it as a major research area in computer vision. Recent advances in NVS, particularly those which use Neural Radiance Fields (NeRF) Mildenhall et al. (2021) and 3D Gaussian splatting (3DGS) Kerbl et al. (2023), have significantly improved NVS performance Gao et al. (2022); Ma & Liu (2018); Zhu et al. (2023). However, these methods heavily rely on dense input views Chibane et al. (2021) and accurate camera poses Jain et al. (2021); Yu et al. (2021b) and typically initialize from sparse point clouds derived from Structure from Motion (SfM) Ullman (1979); Schönberger et al. (2016).

In many real-world applications, obtaining dense views is not feasible, and the available views are usually sparse, covering only a limited number of perspectives Jain et al. (2021); Barron et al. (2021); Yan et al. (2023); Chibane et al. (2021). This sparse-view scenario poses substantial challenges as the large number of unobserved viewpoints results in significant information gaps, which critically impact the completeness and quality of reconstructions. For example, sparse inputs often lack sufficient overlap, which hampers the ability of SfM to estimate camera parameters accurately, causing it to struggle under sparse-view conditions. Particularly in areas of poor texture and smooth surfaces, SfM frequently fails to accurately match features across multiple images. This often results in rendered scenes that are plagued by artifacts and inconsistencies Niemeyer et al. (2022); Chen et al. (2021); Yu et al. (2021b). The fundamental issue here stems from a lack of sufficient prior information. A straightforward approach to address this is to provide the model with more

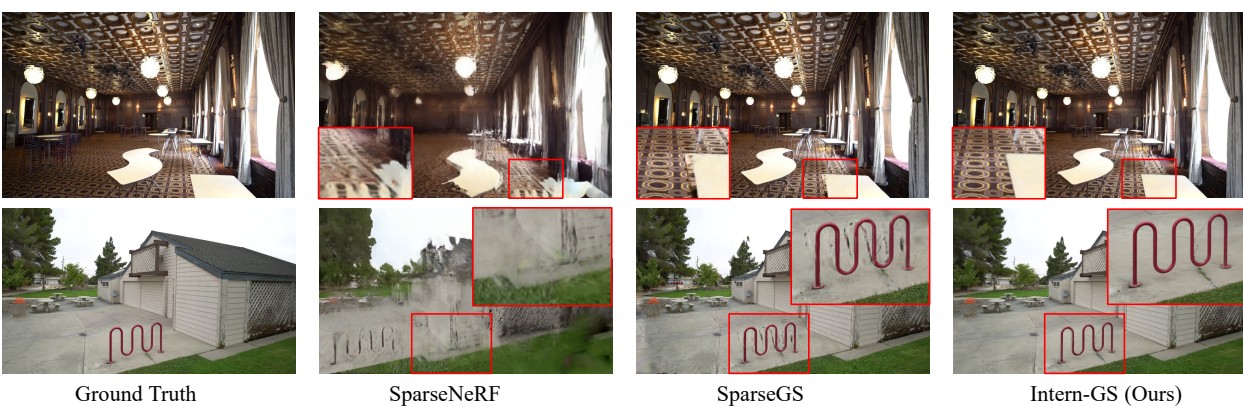

| Ground Truth | SparseNeRF | SparseGS | Intern-GS (Ours) |

Figure 1: Comparison of the SOTA SparseNeRF Wang et al. (2023a), SparseGS Fu et al. (2024b) in 3 training views. Our work leverage multi-view stereo prior to densely initial 3D Gaussian, supervised using a combination of various forms of regularization. From the reconstruction results in the figure, it is evident that our method significantly enhances rendering quality, yielding more refined and detailed results.

accurate and robust prior information. In this context, vision foundation models Riquelme et al. (2021); Radford et al. (2021); Liu et al. (2021), which are pre-trained on extensive and diverse datasets, present a promising avenue by providing comprehensive visual priors that significantly help bridge information gaps in sparse-view NVS.

To effectively address the challenges of sparse-view conditions, we propose **Intern-GS**, a novel approach that leverages priors from vision foundation models to guide the initialization and optimization process of 3D Gaussian Splatting in sparse-view settings. Our key idea is to mitigate information gaps from unobserved views by extracting and integrating priors from vision foundation models, allowing our method to generate dense and redundancy-free Gaussian initializations and refine depth and appearance across unobserved perspectives. Specifically, our method starts with DUSt3R Wang et al. (2023b) first, a state-of-the-art multi-view stereo model, to create a dense point cloud initialization, overcoming the limitations of traditional SfM-based sparse initializations. Unlike conventional SfM techniques relying on feature matching between images, DUSt3R leverages robust multi-view stereo priors to directly learn the mapping between 2D images and 3D point maps. Due to the point-to-point prediction nature of DUST3R, its initialization tends to produce a large number of redundant representations, which can in turn weaken the model's representation capacity. To alleviate this issue, we propose a novel redundancy reduction strategy. Specifically, for the first frame, we initialize a Gaussian for every pixel. For each subsequent frame, we render a confidence map using the Gaussians created from previous frames, and identify under-represented regions as those with confidence values below a predefined threshold. Only these regions are then initialized with new Gaussians. During the optimization process, Intern-GS employs advanced visual models to improve the appearance and depth of unobserved viewpoints. We leverage a pre-trained diffusion model Wang et al. (2024) to enhance appearance by generating realistic textures for areas without direct observation. Concurrently, we use a deep depth estimation model to predict depth values for unobserved viewpoints, providing key geometric constraints that guide the Gaussian optimization toward more accurate 3D geometry across perspectives. This hybrid regularization fully accounts for both color and geometric information from both observed and unobserved views, providing comprehensive and sufficient guidance for Gaussian optimization.

Intern-GS undergoes rigorous evaluation across diverse sparse-view datasets, demonstrating state-of-the-art performance on both forward-facing and large-scale scenes. It surpasses existing methods on challenging benchmarks such as LLFF Mildenhall et al. (2019), DTU Aanæs et al. (2016), and Tanks and TemplesKnapitsch et al. (2017), setting new standards for sparse-view NVS by delivering highly accurate and consistent reconstructions.

Our main contributions are summarized as follows:

- We propose Redundancy Free Initialization (RF), a novel approach for dense and non- Gaussian initialization that leverages multi-view stereo priors from vision foundation models, achieving geometrically

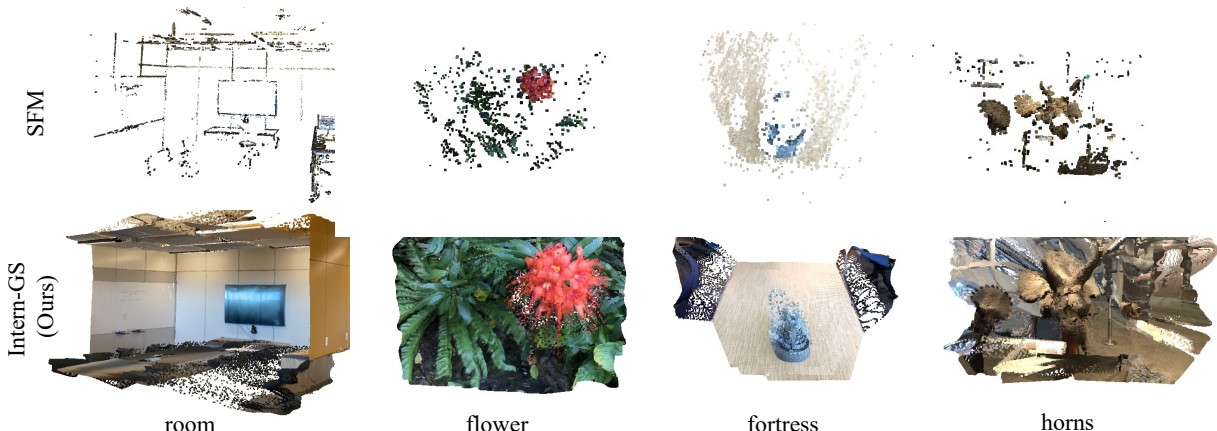

Figure 2: Comparison of point cloud initialization of original 3D Gaussian Kerbl et al. (2023) and our method in 4 scenes under 3 training views. The first row's results are derived from SfM Ullman (1979) used by the original 3D Gaussian and most NeRF-based methods. In contrast, the second row shows the results of our initialization method. Obviously, our method outperforms the SfM method in texture-poor areas.

consistent, concise and efficient initialization that outperforms traditional sparse initialization methods, particularly in low-texture areas and redundant representation regions.

- We introduce a comprehensive hybrid regularization mechanism that leverages diffusion and depth priors to optimize 3D Gaussian representations, ensuring consistent color and geometry across both visible and unobserved viewpoints.

- Extensive experiments validate Intern-GS as a state-of-the-art approach for sparse-view NVS, excelling on challenging datasets such as LLFF, DTU, and Tanks and Temples and setting a new standard for sparse-view novel view synthesis.

## 2 Related Work

### 2.1 Novel View Synthsis

The goal of novel view synthesis Zhou et al. (2016); Avidan & Shashua (1997) is to create images of a scene or object from unseen viewpoints based on images from specific seen viewpoints. Recent technological advancements have shown exciting progress, one of which is Neural Radiance Fields (NeRF) Mildenhall et al. (2021). It employs a multilayer perceptron (MLP) Rumelhart et al. (1986) to map spatial locations and viewing angles to colors and densities, rendering images through volume rendering. Subsequent improvements have primarily focused on enhancing its quality Mildenhall et al. (2021); Barron et al. (2021); Zhang et al. (2020); Deng et al. (2022)and efficiency Yu et al. (2021a); Garbin et al. (2021); Reiser et al. (2021); Chen et al. (2022); Liu et al. (2020) of the renderings, as well as improving 3D generation capabilitiesTang et al. (2023); Chan et al. (2022); Niemeyer & Geiger (2021); Meng et al. (2021); Schwarz et al. (2020) and refining pose estimation techniques Sucar et al. (2021); Rosinol et al. (2023); Zhu et al. (2022).

However, achieving real-time rendering performance remains a significant challenge, as NeRF requires extensive computational resources and processing time. The introduction of 3D Gaussian Splatting Kerbl et al. (2023) addresses this issue, shifting more attention towards this explicit representation. This approach involves initializing a series of anisotropic 3D Gaussian ellipsoids to model the entire radiance field comprehensively, followed by rendering the scene using differentiable splatting methods. This technique has proven highly effective in rapidly and accurately reconstructing complex real-world scenes, delivering robust performance for scenes under multi-view input Tang et al. (2023). Despite these advancements, numerous challenges remain, particularly when dealing with sparse view inputs.

## 2.2 Novel View Synthsis in Sparse-View

Both of these popular methods exhibit significant flaws: they rely on dense input and accurate estimation of camera parameters. To address the issue of reduced accuracy that arises from the diminished matches in corresponding points in Structure from Motion (SfM) as the number of input views decreases, numerous studies have employed strategies that incorporate additional prior information or have designed specific regularization terms to enhance the performance of NeRFJain et al. (2021); Deng et al. (2022); Niemeyer et al. (2022); Wang et al. (2023a); Truong et al. (2023); Wu et al. (2024) and 3DGS Zhu et al. (2024); Fu et al. (2024a); Zhang et al. (2024); Fu et al. (2024b); Fan et al. (2024). For instance, DietNeRF Jain et al. (2021) leverages semantic priors obtained from the large semantic model CLIP to introduce extra semantic constraints in high dimensions, ensuring multi-view consistency of the rendered unseen views. RegNeRF Niemeyer et al. (2022) regularizes the geometry and appearance of patches rendered from unobserved viewpoints and anneals the ray sampling space during training. SparseNeRF Wang et al. (2023a) utilize a local depth ranking method to make sure the expected depth ranking of the NeRF is consistent with that of the coarse depth maps in local patches. Sparf Truong et al. (2023) optimizes NeRF's parameters and noisy poses simultaneously, designing consistency constraints across different viewpoints to address the adverse effects of inaccurate poses on model optimization. Reconfusion Wu et al. (2024) first uses a diffusion prior to impose color regularization constraints on NeRF at unseen viewpoints.

Simultaneously, there have been some improvements based on 3DGS. For instance, FSGS Zhu et al. (2024) was the first to introduce geometric constraints on training viewpoints, utilizing depth information to ensure the newly generated 3DGS is positioned correctly. CF-3DGS Fu et al. (2024a) utilizes the continuity of the input video stream to add 3DGS frame by frame, thereby eliminating the need for SfM. CoR-GS Zhang et al. (2024) employs an unsupervised approach to simultaneously train two 3DGS representations, evaluate their inconsistencies, and collaboratively prune Gaussians that exhibit high point disagreement at inaccurate positions. SparseGS Fu et al. (2024b) combines depth priors, novel depth rendering techniques, and pruning heuristic algorithms to mitigate floating artifacts. However, these methods overlook a critical point: as an explicit representation, 3DGS heavily relies on an accurate, non-redundant initialization.

Recently, with the rapid development of feed-forward models, like Dust3R Wang et al. (2023b) and vision foundation models, many 3DGS-based methods have begun to incorporate them as priors. Representative works include InstantSplat Fan et al. (2024), LM-Gaussian Yu et al. (2024a), and ViewCrafter Yu et al. (2024b). InstantSplat leverages DUST3R for initialization and uses color as a supervision signal while jointly optimizing poses. LM-Gaussian employs a diffusion prior to refine rendered RGB images, enhancing color consistency across unobserved views. ViewCrafter further improves multi-view color consistency by adopting a video diffusion prior. Although these methods explore the use of vision priors to enhance Gaussian quality, they often focus on limited aspects. To address these limitations, we propose a more holistic solution that jointly considers color and geometry consistency in both visible and invisible views.

# 3 Method

In this section, we first introduce the preliminaries of 3D Gaussians 3.1. Then, we introduce our dense and non-redundant Gaussian initialization method 3.2. Finally, We present our proposed hybrid regularization strategy 3.3 and 3.4.

## 3.1 Preliminary of 3D Gaussian Splatting

**Representation.** The process of 3D scene representation typically begins with the extraction of a 3D point cloud from a set of input images, which is commonly achieved through SfM techniques Schonberger & Frahm (2016). 3D Gaussian Splatting initializes each point cloud into a corresponding 3D Gaussian $G(x)$. Each Gaussian $G_i(x)$ includes the following attributes: opacity $\alpha_i$ and color $c_i$, where the color is defined by a $l$ dimensional spherical harmonic function $\{c_i \in \mathbb{R}^3 \mid i = 1, 2, \dots, l^2\}$. In 3D space, the position and shape of each Gaussian distribution are defined by its mean (parameters describing its center position) $\mu_i$ and covariance matrix $\Sigma_i$ (parameters describing its spread and shape). The representation of the $i$-th Gaussian

$G_i(x)$ is given by:

$$G_i(x) = e^{-\frac{1}{2}(x-\mu_i)^T \Sigma_i^{-1}(x-\mu_i)}, \tag{1}$$

where the covariance matrix $\Sigma_i$ is calculated from the scale $s_i$ and rotation $r_i$, as $\Sigma = RSS^T R^T$, where $R$ is a matrix of quaternions representing rotation, and $S$ is a matrix representing scaling. Overall, the optimizable parameters of the $i$-th Gaussian $G_i(x)$ are $\{\alpha_i, c_i, \mu_i, r_i, s_i\}$ .

**Rendering and Optimization Process.** To calculate the color of each pixel in the rendered 2D image, 3D Gaussian Splatting employs a rasterization process, combining contributions from $N$ Gaussians impacting that pixel to synthesize the color $C_p$ of pixel $p$:

$$C_p = \sum_{i \in N} c_i \alpha_i' \prod_{j=1}^{i-1}(1 - \alpha_j'), \tag{2}$$

where $c_i$ represents the color contribution of the $i$-th Gaussian, computed from its spherical harmonics (SH) coefficients, and $\alpha_i'$ denotes the effective opacity of the $i$-th Gaussian in the 2D-pixel coordinate system. The effective opacity $\alpha'$ is derived from the projected covariance matrix $\Sigma'$ in the 2D plane and the original opacity $\alpha$ of the 3D Gaussian. The projection of the 3D Gaussian into the 2D pixel coordinate system is achieved through a transformation:

$$\Sigma' = JW\Sigma W^T J^T, \tag{3}$$

where $J$ is the approximate Jacobian matrix of the projection transformation, and $W$ represents the rotational component of the camera pose. We use a similar method to render depth, as described in the following formula:

$$D_p = \sum_{i \in N} d_i \alpha_i' \prod_{j=1}^{i-1}(1 - \alpha_j'), \tag{4}$$

where $d_i$ represents the depth at each pixels, which is calculate by the distance to the camera center $o$, the formula is

$$d_i = \|\mu_i - o\|_2. \tag{5}$$

3D Gaussian Splatting optimizes its model parameters through color constraints. During the optimization process, the algorithm dynamically adapts the Gaussian distributions by cloning or splitting them based on their gradient magnitudes and scales. Specifically, it clones Gaussians whose gradients exceed a predefined threshold but whose scales remain below a certain limit. At the same time, it splits Gaussians with gradients and scales, both exceeding their respective thresholds. This adaptive optimization strategy ensures a balance between detail preservation and computational efficiency. In our work, we retain the original optimization methodology and color constraints.

### 3.2 Multi-View Stereo Guided Dense Initialization

Existing initialization techniques, such as Structure from Motion usually generate sparse correspondences that lack sufficient geometric details and make it difficult to fully exploit color priors. In addition, SfM-based methods are computationally intensive, limiting their feasibility in real-time applications. This phenomenon highlights the urgent need to design more efficient and robust initialization strategies for 3D Gaussian Splatting, especially for applications in real-time high-fidelity scene reconstruction.

**Redundancy-Free Initialization.** With globally aligned points, poses, and pixel-level confidence maps, we initially generate a dense point cloud. However, this introduces significant redundancy, where multiple points occupy the same spatial location, leading to inefficiencies in storage and computation. To address this, we propose a Redundancy-Free (RF) Strategy to downsample the point cloud, reducing redundancy while preserving essential scene details and improving representation efficiency. Inspired by Matsuki et al. (2023); Keetha et al. (2023), we randomly select a primary viewpoint and initialize new Gaussians using all its pixels. Since some regions are already well-represented, indiscriminately adding Gaussians would cause

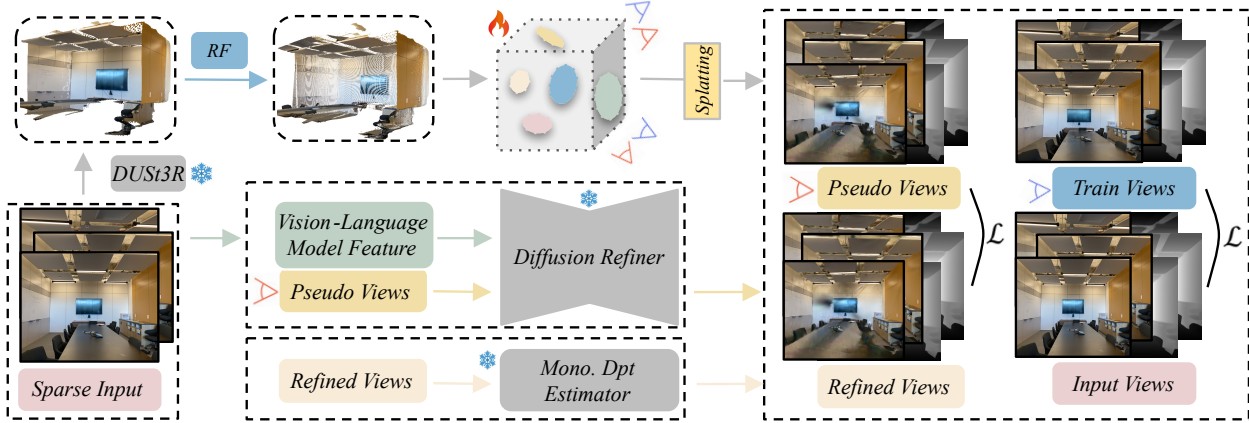

Figure 3: In our framework, we first utilize a multi-view stereo to predict point maps. This technique recovers point maps at a consistent scale, but failed to represent scene because of redundancy in points. To handle overlapping regions in the point maps, we designed a Redundancy-Free (RF) algorithm that only initializes areas which have not been well defined for all views. For the optimization progress, we design a novel regularization method that jointly constrains the depth and color information of training and pseudo views. The color supervision is derived from the diffusion refine model we employ, while the depth supervision comes from multi-view stereo model and monocular depth prediction model.

unnecessary duplication. To prevent this, we introduce a masking mechanism to selectively determine where new Gaussians should be introduced. The mask formulation is as follows:

$$M_p = (S_p < 0.5) + \left(D_p^{GT} < D_p\right)(L_1(D_p) > 50MDE), \tag{6}$$

where $p$ represents each pixel, $S_p$ represents the track, indicating the density of the Gaussian at that point, which is calculated similar to color and depth as:

$$S_p = \sum_{i \in N} s_i \alpha_i' \prod_{j=1}^{i-1}(1 - \alpha_j'), \tag{7}$$

where $s_i$ represents the Gaussian weight, and $\alpha_i'$ is the opacity of the $i$-th Gaussian in the accumulation process. The mask $M_p$ ensures that new Gaussians are only added in regions where the density is insufficient or where the estimated depth $D_p$ is positioned in front of the ground truth depth $D_p^{GT}$, and the depth error exceeds 50 times the median depth error (MDE). The addition of new Gaussians follows the same initialization procedure as that used for the primary viewpoint.

### 3.3 Depth Regularization

In sparse-view settings, optimizing Gaussians solely with multi-view photometric loss is not that adequate, as it constrains appearance without fostering a coherent geometric structure. To address this issue, we introduce additional priors and regularization terms. Depth priors derived from pre-trained multi-view stereo will intuitively guide the model toward the correct geometric structure.

**Training Depth Regularization.** Our multi-view stereo depth prior originates from the pre-trained model DUSt3R Wang et al. (2023b), which provides relative depth. To address the scale ambiguity between real-world scenes and estimated depths, we employ a relaxed relative loss method, *Pearson* correlation coefficient, formulated as follows:

$$\text{Corr}(D_{\text{ras}}, D_{\text{est}}) = \frac{\text{Cov}(D_{\text{ras}}, D_{\text{est}})}{\sqrt{\text{Var}(D_{\text{ras}})\text{Var}(D_{\text{est}})}}. \tag{8}$$

Here, $D_{\text{est}}$ is the depth estimated by the multi-view stereo model, and $D_{\text{ras}}$ is the depth rendered by our system. This constraint benefits from being unaffected by scale inconsistencies, optimizing the correlation between the two depths.

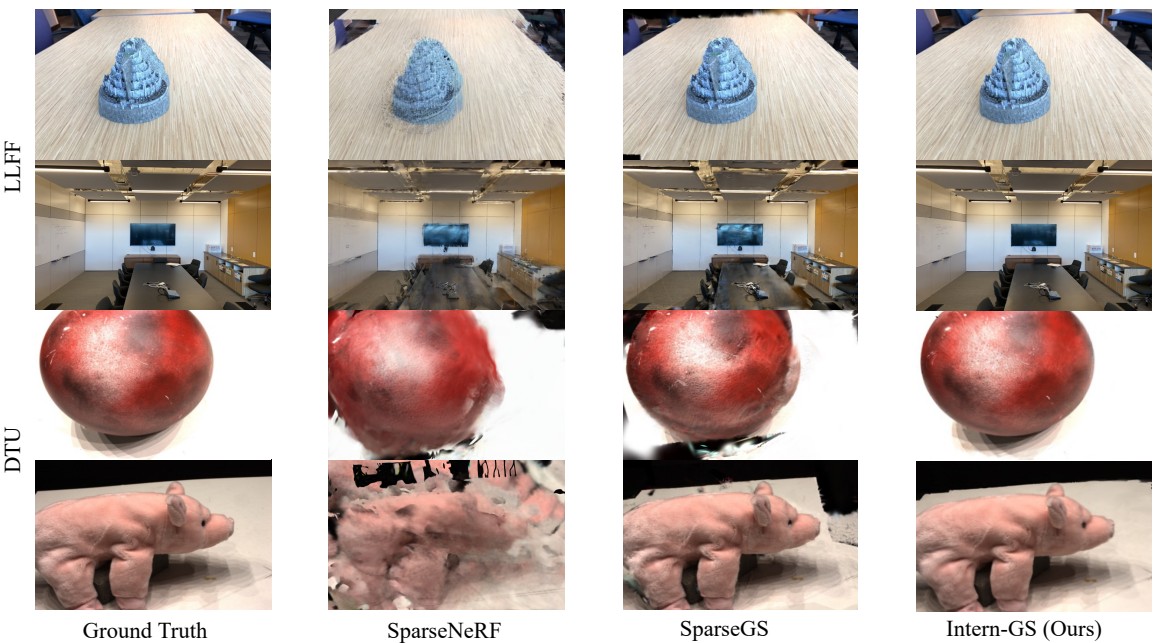

Figure 4: Results on LLFF dataset Mildenhall et al. (2019) and DTU dataset Aanæs et al. (2016) in 3 training views. Our method captures more scene details, particularly in areas with sparse texture information. The SparseNeRF Wang et al. (2023a) approach struggles to synthesize accurate new views under sparse viewpoints, while SparseGS Fu et al. (2024b) produces overly smooth views, losing many details.

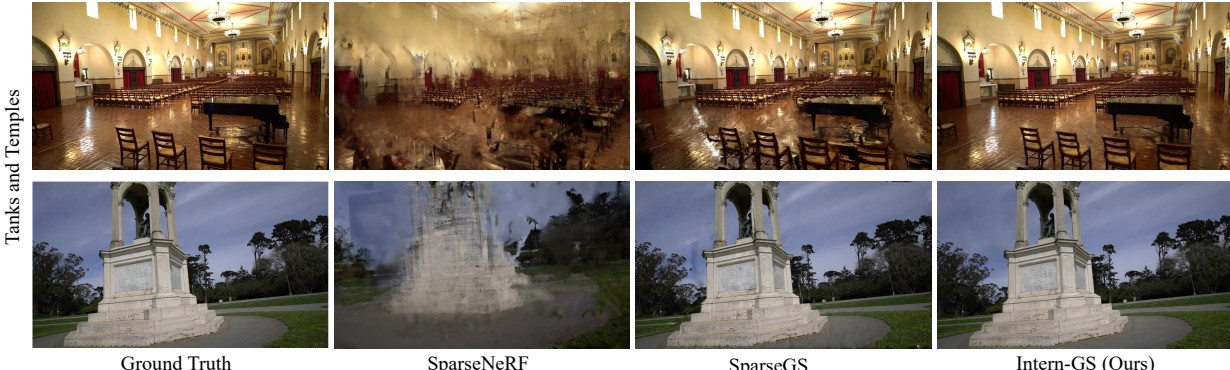

Figure 5: Results on Tanks dataset Knapitsch et al. (2017) in 3 training views. In comparison, SparseNeRF Wang et al. (2023a) struggles to accurately represent structures. While SparseGS Fu et al. (2024b) performs well overall, it tends to lose some texture information in areas with flat depth. In contrast, Intern-GS effectively captures these texture details.

**Pseudo Depth Regularization.**    To improve the generalization of 3D Gaussian across unseen views and reduce overfitting, we first set up pseudo views with a 5-degree deviation on the rotation matrix according to the training viewpoints, and then similarly establish depth constraints for these pseudo views. The equation is as follows:

$$\text{Corr}(D_{\text{ras}}^{pse}, D_{\text{est}}^{pse}) = \frac{\text{Cov}(D_{\text{ras}}^{pse}, D_{\text{est}}^{pse})}{\sqrt{\text{Var}(D_{\text{ras}}^{pse})\text{Var}(D_{\text{est}}^{pse})}}. \tag{9}$$

Here, $D_{\text{ras}}^{pse}$ represents the depth rendered from the pseudo viewpoint, while $D_{\text{est}}^{pse}$ is the depth predicted by the monocular depth pre-trained model based on the RGB image rendered and then refined by Multi-view Appearance Refinement model from the same viewpoint.

### 3.4 Multi-view Appearance Refinement

Above, we improve 3D Gaussian representation by ensuring geometry consistency with added depth regularization. However, color inconsistencies persist in images generated from unseen views, leading to a noisy representation. To address this, we've developed a Multi-view Appearance Refinement (MAR) algorithm. It begins by rendering $N$ images from the Gaussian field at predefined viewpoints, denoted as $I_n$ for $n \in (1, N)$. These rendered images are then refined using diffusion models to produce new images $\hat{I}_n$, , which leverage photometric priors to optimize for correct color consistency.

**Diffusion Process.** For training, starting with the images $I_0$, we initially employ the forward process of the diffusion model to introduce noise, resulting in a set of noisy images, $I_t$, where $t$ represents a specific time step. The equation is as follows:

$$I_t = I_0 + \sigma_t^2 \epsilon, \quad \epsilon \sim \mathcal{N}(\mathbf{0}, \mathbf{I}), \tag{10}$$

where $log\sigma_t \sim \mathcal{N}(P_{\text{mean}}, P_{\text{std}}^2)$ Karras et al. (2022) and $P_{\text{mean}}$ was set to 1.5 $P_{\text{std}}$ was set to 2.0. In the reverse process, diffusion model denoises $I_t$ with a learnable UNet $\mathcal{U}_\theta$:

$$\hat{I}_0 = \mathcal{U}_\theta(I_t; \sigma_t, \mathbf{y}, \mathbf{Z}), \tag{11}$$

where $\mathbf{y}$ represents the semantic-level latent obtained from CLIP, and $\mathbf{Z}$ represents the pixel-level latent obtained from DINO.

For the sampling progress, the images $I_0$ is restored from a randomly-sampled Gaussian noise $I_t$ conditioning on image latent prompt $\mathbf{Z}$ and semantic latent prompt $\mathbf{y}$ by iteratively applying the denoising process with trained UNet $\mathcal{U}$. The equation is as follows:

$$I_T \sim \mathcal{N}(\mathbf{0}, \sigma_T^2 \mathbf{I}), \tag{12}$$

$$I_{t-1} = \frac{I_t - \mathcal{U}_\theta(I_T; \sigma_t, \mathbf{y}, \mathbf{Z})}{\sigma_t}(\sigma_{t-1} - \sigma_t) + I_t, \quad 0 < t \le T \tag{13}$$

where $\sigma_0, \cdots, \sigma_T$ are sampled from a fixed variance schedule of a denoising process with $T$ steps.

**Photometric Consistently Regularization.** After obtaining images from the pseudo viewpoints generated by the diffusion model, we use the loss of color the same as 3D Gaussian Splatting to calculate the loss between the rendered images and the diffusion-generated images,

$$L_{\text{cp}} = L_1\left(I_0, I_0^{'}\right) + \lambda^{'} L_{\text{D-SSIM}}\left(I_0, I_0^{'}\right), \tag{14}$$

where $I_0$ represents the rendered images, $I_0^{'}$ represents the diffusion-generated images.

### 3.5 Hybrid regularization

**Loss Function.** As outlined above, the loss function is consists of four main components: color regularization loss $L_c$, depth regularization loss for training views $L_d$, depth regularization loss for pseudo views $L_{dp}$, and color regularization loss for pseudo views $L_{cp}$. The formula is as follows:

$$L = \lambda_1 L_{\text{c}} + \lambda_2 L_{\text{d}} + \lambda_3 L_{\text{dp}} + \lambda_4 L_{\text{cp}}. \tag{15}$$

The calculation of $L_c$ based on the original 3D Gaussian is as follows:

$$L_{\text{c}} = L_1(\hat{I}, I) + \lambda' L_{\text{D-SSIM}}(\hat{I}, I), \tag{16}$$

where $\hat{I}$ represents the rendered images, $I$ represents the ground-truth images. We set $\lambda_1$, $\lambda_2$ as 0.5, 1 the same as baseline Zhu et al. (2024) and $\lambda_3$, $\lambda_4$ as 0.05, 0.001 respectively due to grid search and $\lambda'$ as 0.2 due to 3D Gaussian Splatting Kerbl et al. (2023).

Table 1: Comparison of PSNR, LPIPS Wang et al. (2004), and SSIM Zhang et al. (2018) with current baseline methods for the novel view synthesis task on the LLFF Mildenhall et al. (2019) and DTU Aanæs et al. (2016) datasets. Some baseline results in the table are sourced from Wang et al. (2023a), and the state-of-the-art results are highlighted in bold black.

| Methods | LLFF | | | DTU | | |
|---|---|---|---|---|---|---|
| | PSNR↑ | LPIPS↓ | SSIM↑ | PSNR↑ | LPIPS↓ | SSIM↑ |
| PixelNeRF Yu et al. (2021b) | 7.93 | 0.682 | 0.272 | 16.82 | 0.270 | 0.695 |
| RegNeRF Niemeyer et al. (2022) | 19.08 | 0.336 | 0.587 | 18.89 | 0.190 | 0.695 |
| FreeNeRF Yang et al. (2023) | 19.63 | 0.308 | 0.612 | 19.92 | 0.182 | 0.787 |
| SparseNeRF Wang et al. (2023a) | 19.86 | 0.328 | 0.624 | 19.55 | 0.201 | 0.769 |
| 3DGS Kerbl et al. (2023) | 15.52 | 0.405 | 0.408 | 10.99 | 0.313 | 0.585 |
| FSGS Zhu et al. (2024) | 20.31 | 0.288 | 0.652 | 19.54 | 0.199 | 0.732 |
| DNGaussian Li et al. (2024) | 19.12 | 0.294 | 0.591 | 18.91 | 0.176 | 0.790 |
| SparseGS Zhu et al. (2024) | 19.86 | 0.322 | 0.668 | 18.89 | 0.178 | 0.834 |
| InstantSplat Fan et al. (2024) | 17.67 | 0.379 | 0.603 | 17.55 | 0.212 | 0.634 |
| LM-Gaussian Yu et al. (2024a) | 19.63 | 0.228 | 0.614 | 18.74 | 0.223 | 0.812 |
| ViewCrafter Yu et al. (2024b) | 19.49 | 0.250 | 0.649 | 19.46 | 0.182 | 0.810 |
| **Intern − GS(Ours)** | **20.49** | **0.212** | **0.693** | **20.34** | **0.163** | **0.851** |

Table 2: Comparison of PSNR, LPIPS Wang et al. (2004), and SSIM Zhang et al. (2018) with state-of-the-art (SOTA) methods on the Tanks and Temples dataset Knapitsch et al. (2017). The state-of-the-art results are highlighted in bold black.

| Method | PSNR↑ | LPIPS↓ | SSIM↑ |
|---|---|---|---|
| PixelNeRF Yu et al. (2021b) | 11.93 | 0.598 | 0.248 |
| RegNeRF Niemeyer et al. (2022) | 19.64 | 0.243 | 0.718 |
| FreeNeRF Yang et al. (2023) | 20.82 | 0.210 | 0.729 |
| SparseNeRF Wang et al. (2023a) | 21.98 | 0.219 | 0.730 |
| 3DGS Kerbl et al. (2023) | 15.36 | 0.379 | 0.572 |
| FSGS Zhu et al. (2024) | 22.31 | 0.197 | 0.693 |
| DNGaussian Li et al. (2024) | 20.69 | 0.277 | 0.721 |
| SparseGS Zhu et al. (2024) | 21.20 | 0.231 | 0.717 |
| InstantSplat Fan et al. (2024) | 22.20 | 0.199 | **0.743** |
| LM-Gaussian Yu et al. (2024a) | 21.34 | 0.219 | 0.705 |
| ViewCrafter Yu et al. (2024b) | 22.12 | 0.208 | 0.718 |
| Intern-GS (Ours) | **22.67** | **0.191** | 0.736 |

# 4 Experiments

In this section, we will elaborate on the experimental setup, including the datasets used and the comparison methods. Additionally, we will present the experimental evaluation results and conduct comprehensive ablation studies to validate the effectiveness of each component.

## 4.1 Experimental Settings

**Datasets.** We conduct our experiments on three widely used datasets, LLFF Mildenhall et al. (2019), DTU Aanæs et al. (2016), and Tanks and Temples Knapitsch et al. (2017), to comprehensively evaluate our method across diverse scene types. For the LLFF dataset, we follow previous works Wang et al. (2023a); Yang et al. (2023) by splitting the images into three designated training views and multiple test views. For the DTU dataset, we adopt the experimental protocol used in SPARF Truong et al. (2023) and RegNeRF Niemeyer

Table 3: Ablation study on LLFF Mildenhall et al. (2019) with three training views, analyzing the individual contributions of the proposed three modules: Multi-View Stereo Guided Dense Initialization, Depth Regularization, and Multi-view Appearance Refinement. The results demonstrate that Multi-View Stereo Guided Dense Initialization has the most significant impact on the experimental outcomes.

| Dense Init. (3.2) | Depth Regu. (3.3) | MAR (3.4) | PSNR ↑ | LPIPS ↓ | SSIM ↑ |
|---|---|---|---|---|---|
| ✗ | ✗ | ✗ | 15.52 | 0.405 | 0.408 |
| ✓ | ✗ | ✗ | 19.21 | 0.341 | 0.573 |
| ✓ | ✓ | ✗ | 19.64 | 0.329 | 0.640 |
| ✓ | ✓ | ✓ | **20.49** | **0.304** | **0.656** |

et al. (2022), training our model on the same three training views and evaluating it on the corresponding test views. To eliminate background noise and focus on the target object, we use the same object masks during evaluation. To further assess the model's applicability to non-forward-facing scenes, we conduct additional experiments on the Tanks and Temples dataset, following the approach outlined in InstantSplat Fan et al. (2024). We apply downsampling rates of 8 and 4 for LLFF and DTU, and none downsampling for Tanks and Temples dataset.

**Comparison Methods.** Following the previous neural radiance fields in few-shot setting, we compare our Intern-GS with current high performing methods, including PixelNeRF Yu et al. (2021b), RegNeRF Niemeyer et al. (2022), FreeNeRF Yang et al. (2023), and SparseNeRF Wang et al. (2023a). We also report the results of raw 3D Gaussian Splatting Kerbl et al. (2023), FSGS Zhu et al. (2024), DNGaussian Li et al. (2024), SparseGS Fu et al. (2024b), the similarly pose-free InstantSplat Fan et al. (2024), LM-Gaussian Yu et al. (2024a) and ViewCrafter Yu et al. (2024b). The results of some of previous works are reported directly from the respective published papers for comparisons and for baselines that were not reported in previous work, we deployed and replicated the same fair experiments, and presented the results. To quantitatively evaluate the reconstruction performance, we employ Peak Signal-to-Noise Ratio (PSNR), Structural Similarity Index (SSIM) Zhang et al. (2018), and Learned Perceptual Image Patch Similarity (LPIPS) Wang et al. (2004) as evaluation metrics across all methods, providing a comprehensive assessment of both accuracy and perceptual quality.

## 4.2 Comparison to Baseline

**LLFF Datasets.** Intern-GS has been comprehensively evaluated on the LLFF dataset, demonstrating robust performance across both qualitative and quantitative assessments. As shown in Figure 4 and Table 1, our method consistently outperforms all baseline approaches across all evaluation metrics. Statistically, Intern-GS achieves higher accuracy and fidelity, highlighting its effectiveness in reconstructing detailed and structurally consistent 3D representations.

From a visual perspective, our model excels in capturing fine-grained geometric details, particularly in challenging regions with sparse texture information. In contrast, SparseNeRF Wang et al. (2023a) encounters difficulties under sparse viewpoints, leading to incomplete or less reliable reconstructions. Although SparseGS Fu et al. (2024b) incorporates monocular depth as an additional constraint, its reliance on sparse point cloud initialization results in lower geometric completeness compared to our method, which benefits from a dense initialization strategy. By leveraging a more robust depth-aware approach, Intern-GS effectively reconstructs complex structures with higher precision, making it particularly well-suited for real-world applications requiring high-quality 3D scene understanding.

**DTU Datasets.** We present the results of our method on the DTU dataset in Figure 4 and Table 1. Unlike the LLFF dataset, the DTU dataset is characterized by a prominent central object against a black background. This distinction necessitates a careful evaluation process when computing metrics for novel view synthesis. Specifically, following the baseline approach, we apply a corresponding mask to each viewpoint and

Table 4: Ablation study of Redundancy-Free Initialization on LLFF Mildenhall et al. (2019) with three training views.

| DUSt3R | RF. (3.2) | PSNR ↑ | LPIPS ↓ | SSIM ↑ | GS Numbers |
|:---:|:---:|:---:|:---:|:---:|:---:|
| ✗ | ✗ | 15.52 | 0.405 | 0.408 | 4622 |
| ✓ | ✗ | 18.48 | 0.348 | 0.521 | 571536 |
| ✓ | ✓ | **19.21** | **0.341** | **0.573** | 196165 |

Table 5: Ablation study of different depth regularization type on LLFF Mildenhall et al. (2019) with three training views.

| Train View | Pseudo View(3.2) | PSNR ↑ | LPIPS ↓ | SSIM ↑ |
|:---:|:---:|:---:|:---:|:---:|
| ✗ | ✗ | 19.21 | 0.341 | 0.573 |
| ✓ | ✗ | 19.51 | 0.331 | 0.620 |
| ✗ | ✓ | 19.36 | 0.338 | 0.591 |
| ✓ | ✓ | **19.64** | **0.329** | **0.640** |

calculate evaluation metrics only within the masked image regions. This ensures that the black background does not interfere with the overall Gaussian optimization process, leading to a more accurate assessment of reconstruction quality.

Our results demonstrate that Intern-GS consistently achieves superior performance in terms of PSNR, LPIPS, and SSIM compared to previous methods. SparseGS, in particular, struggles with the dataset's noisy and smooth backgrounds, which adversely affect monocular depth estimation and, consequently, the quality of the reconstructed scene. Visually, our approach excels in handling low-texture and smooth regions, where it provides richer depth and color information. This enables our model to capture more precise structural and photometric details, significantly enhancing both the geometric fidelity and the overall visual realism of the synthesized views.

**Tanks and Temples Datasets.** We conducted experiments on the Tanks and Temples dataset, which, unlike the first two datasets, contains large-scale scenes covering a wide range of viewpoints. As shown in Table 2 and Figure 5, our Intern-GS achieved the best LPIPS and PSNR scores and second best in SSIM scores. This result showcases our model's capability with large scenes from non-frontal perspectives. We attribute this to the nature of our diffusion-based prior, which focuses on perceptual quality and color consistency across wide viewpoints. While this benefits large-scale scenes, slight structural deviations in high-frequency regions may lead to a slightly lower SSIM.

### 4.3 Ablation Studies

We conduct ablation studies on three key aspects of our method: 3.2 a comparison between dense initialization and the conventional Structure-from-Motion-based initialization used in 3D Gaussian Splatting, 3.3 the impact of our proposed depth regularization on both training views and 3.4 novel synthesized views, and the effect of diffusion-prior-based appearance refinement in novel synthesized views. Table 3 presents the results of our ablation study on the LLFF dataset Mildenhall et al. (2019) using three training views as a case study.

**Dense Initialization.** We conducted a comparative analysis between our model with Dense and Non-redundancy Initialization and a baseline model without it. As shown in the first and second rows of Table 3, our approach consistently outperforms the original 3D Gaussian Splatting (3DGS) across all three evaluation metrics. This demonstrates that the Gaussian initialization generated via multi-view stereo provides a more reliable geometric prior, particularly in regions with sparse texture information. Furthermore, our redundancy reduction strategy effectively mitigates the influence of low-confidence areas, such as the sky, thereby enhancing the robustness of the initialization process.

**Depth Regularization.** We conducted a comparative study to evaluate the impact of our Depth Regularization by assessing models with and without it. As shown in the third row of Table 3, incorporating depth priors as constraints effectively guides Gaussian optimization toward more accurate geometric representations, enabling the model to learn more coherent and structurally faithful surfaces. This enhancement leads to a 0.43 increase in PSNR, a 0.002 reduction in LPIPS, and a 0.067 improvement in SSIM, demonstrating the effectiveness of our depth-aware regularization in refining scene reconstruction quality.

**Multi-view Appearance Refinement.** We conducted a comparative analysis to assess the impact of our Multi-view Appearance Refinement algorithm by evaluating models with and without it. In novel viewpoint regions, the absence of sufficient initialization points makes it challenging for Gaussians to be effectively optimized, leading to suboptimal representation of these areas. To address this, we introduce diffusion-based refinement from pseudo-views, enabling the model to better optimize from previously unseen perspectives. As shown in Table 3, the consistent improvement across all three evaluation metrics validates the effectiveness of our approach in enhancing appearance fidelity and overall reconstruction quality.

### 4.4 More Ablation Studies

**Redundancy-Free Initialization.** We conducted an additional ablation study on the Redundancy-Free Initialization proposed for the initialization stage. The first row in Table 4 shows the reconstruction results using the original DUSt3R initialization, while the second row presents the results using the original DUSt3R initialization combined with our Redundancy-Free method. In terms of quantitative metrics, our method achieves improvements in PSNR, LPIPS, and SSIM. Furthermore, our initialization approach effectively reduces the number of initial Gaussians, thereby decreasing the required optimization time and improving efficiency.

**Training and Pseudo Depth Regularization.** To further ablate the effects of the depth regularization constraints from the training views and the pseudo views, we conducted an ablation study on both. Our experiments are based on the dense initialization setting. The first row shows the reconstruction results without any depth regularization constraints. The second and third rows show the results when training depth constraint and the pseudo depth constraint is applied, respectively. The fourth row presents the results when both depth regularization constraints are used together. As shown in Table 5, the depth constraint from the training views contributes more significantly to the overall reconstruction performance. However, the depth constraint from the pseudo views also benefits the training process, and the two constraints are compatible and can be effectively combined.

## 5 Conclusion

In this work, we introduce Intern-GS, a novel view synthesis method in sparse-view using dense point cloud initialization. Utilizing multi-view stereo priors, we perform effective, non-redundant initialization in texture-sparse regions. During Gaussian optimization, we use depth priors from multi-view stereo and monocular depth models to guide the model towards the correct geometry. To capture richer color details, especially in unseen areas, and avoids color misinterpolation caused by initialization gaps due to lack of perspective, we leverage diffusion model to refine the pseudo view images. It provides the Gaussian with the missing color guidance under pseudo-viewpoints, helping it to achieve color-consistent optimization in unseen areas. Intern-GS offers a novel solution to the challenge of poor rendering in texture-sparse regions.

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

# A    Appendix

We provide additional materials for our submission. The content is organized as follows:

- In appendix B, we present the additional Implementation Details details of Intern-GS;

- In appendix C, we show the Additional Experimental Results of Intern-GS;

- In appendix D, we discuss the Limitations of our work and the potential Future Work.

# B    Implementation Details

### B.1    Datasets Details

We use three datasets for the experiments, which are the LLFF dataset Mildenhall et al. (2019), the DTU dataset Aanæs et al. (2016), Tanks and Temples Knapitsch et al. (2017). For the LLFF dataset, it includes 8 scenes and the original image size is 4032 * 3024. We use images that have been downsampled by a factor of 8 to 504 * 378 as the input. Regarding the DTU dataset, it has 15 scenes and the original image size is 1600*1200. We resize the image to 400 * 300 as the input. We do not do any downsampling to Tanks and Temples with size 960*540, which has 8 scenes in it.

### B.2    Initialization Details

First, we obtain 3D point maps of the same size as the sparse input images by DUSt3R Wang et al. (2023b). Next, we use global alignment method to align these point maps to the same coordinate system. From this, we obtain point maps, depth information, camera poses and confidence maps consistent in scale. However, to obtain non-redundant points for initialization and inspired by SLAM Keetha et al. (2023), we design a

Table 6: Comparison of PSNR, LPIPS Wang et al. (2004), and SSIM Zhang et al. (2018) with state-of-the-art (SOTA) methods on the Tanks and Temples dataset Knapitsch et al. (2017) in 6 views and 12 views. The state-of-the-art results are highlighted in bold black.

| | PSNR↑ | | SSIM↑ | | LPIPS↓ | |
|---|---|---|---|---|---|---|
| | 6-view | 12-view | 6-view | 12-view | 6-view | 12-view |
| 3DGS Kerbl et al. (2023) | 17.78 | 18.92 | 0.593 | 0.621 | 0.349 | 0.317 |
| FSGS Zhu et al. (2024) | 24.41 | 27.87 | 0.768 | 0.861 | 0.153 | 0.129 |
| DNGaussian Li et al. (2024) | 23.34 | 25.79 | 0.783 | 0.814 | 0.258 | 0.219 |
| SparseGS Zhu et al. (2024) | 23.56 | 26.04 | 0.792 | 0.831 | 0.229 | 0.187 |
| InstantSplat Fan et al. (2024) | 25.45 | 28.19 | 0.8453 | 0.8785 | 0.1173 | **0.1068** |
| LM-Gaussian Yu et al. (2024a) | 25.68 | 28.34 | 0.851 | 0.881 | 0.117 | 0.113 |
| ViewCrafter  Yu et al. (2024b) | 25.62 | 28.23 | 0.832 | 0.877 | 0.121 | 0.118 |
| Intern-GS(Ours) | **26.41** | **28.59** | **0.863** | **0.882** | **0.112** | 0.111 |

method to incrementally add initialization points on a per-image basis. Specifically, for the first image, we initialize all pixel points. For subsequent images, we only initialize areas with an uncertainty less than 0.5. Finally, we obtain scale-aligned and non-redundant initial point clouds for Gaussian initialization.

### B.3 Training Details

During all the 10000 training epoch, the learning rates for position, Spherical Harmonics (SH) coefficients, opacity, scaling, and rotation are set to 0.00016, 0.0025, 0.05, 0.005 and 0.001 respectively. We begin with SH degree of 0 for basic color representation and increment it by 1 every 500 iterations until reaching a max degree of 4, gradually increasing the complexity of representation and we reset the opacity for all Gaussians to 0.05 at iterations 2000, 5000 and 7000 aligned with the original 3D Gaussian splatting. During optimization, we densify every 100 iterations starting from the 500th iteration, continuing for 10000 steps per dataset. We apply color L1 losses one from comparing the rgb images from original training views and rendered training rgb images from the same views, another from comparing the rgb images rendered from the predefined pseudo views and diffusion-refined color images by these images. We also apply *Pearson Correlation Coefficient* losses for depth one from comparing the original depth images from training views gained by DUSt3R and depth images rendered from the same training views, another from comparing depth images from the same predefined pseudo views and depth images estimated by pre-trained monocular depth estimation model, MiDaS Ranftl et al. (2020). We apply two color losses and train view depth loss from the beginning till the end and pseudo-view depth loss from the 2000th iteration. Using an NVIDIA 4090 GPU with 24GB of video memory, we train each scene of the LLFF and DTU datasets for 2-3 minutes, and each scene of the Tanks and Temples dataset for 4-5 minutes.

## C Additional Experimental Results

### C.1 More Visual Results

We present additional qualitative results of Intern-GS on LLFF dataset, DTU dataset and Tanks and Temples dataset. The experimental results are shown in Figure 9, Figure 10, Figure 11, Figure 12. Additionally, we provide the video results of all the dataset in 'video results' folder.

### C.2 6 Views and 12 Views Results

To further improve the completeness of our experiments, we conducted evaluations using sparse 6-view and 12-view inputs. As shown in Table 6, Figure 6, Figure 7, Figure 8 the results demonstrate that our method outperforms all baselines under the 6-view setting, while the advantage becomes less significant under the

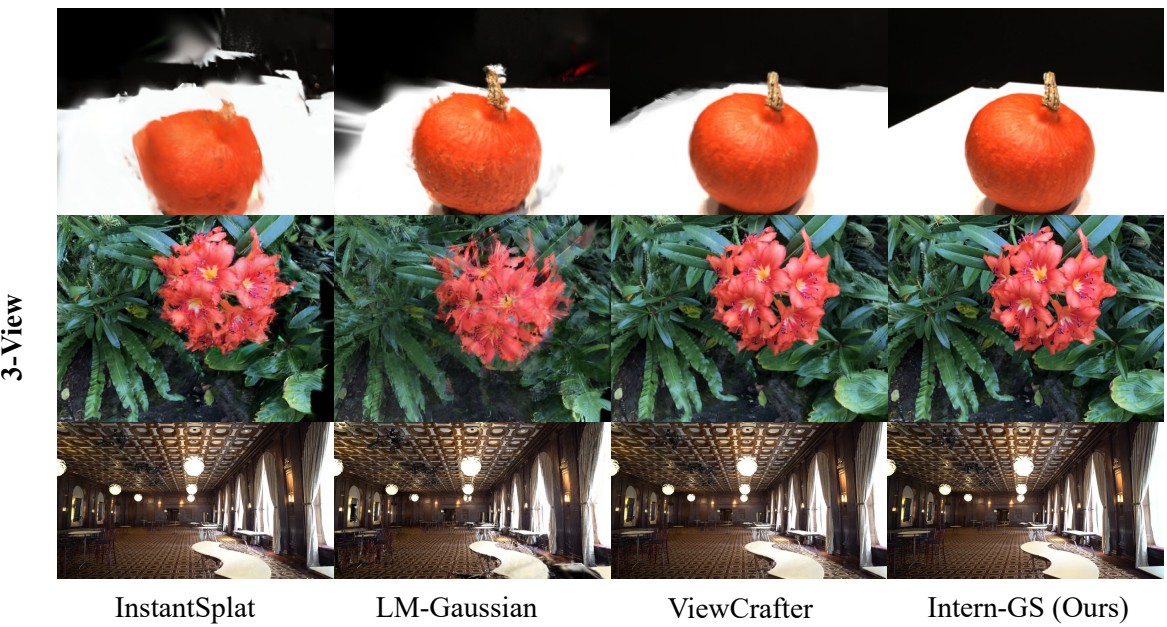

**3-View**

InstantSplat  LM-Gaussian  ViewCrafter  Intern-GS (Ours)

Figure 6: The comparison results between Instantsplat, LM-Gaussian, ViewCrafter and our method in 3-view input.

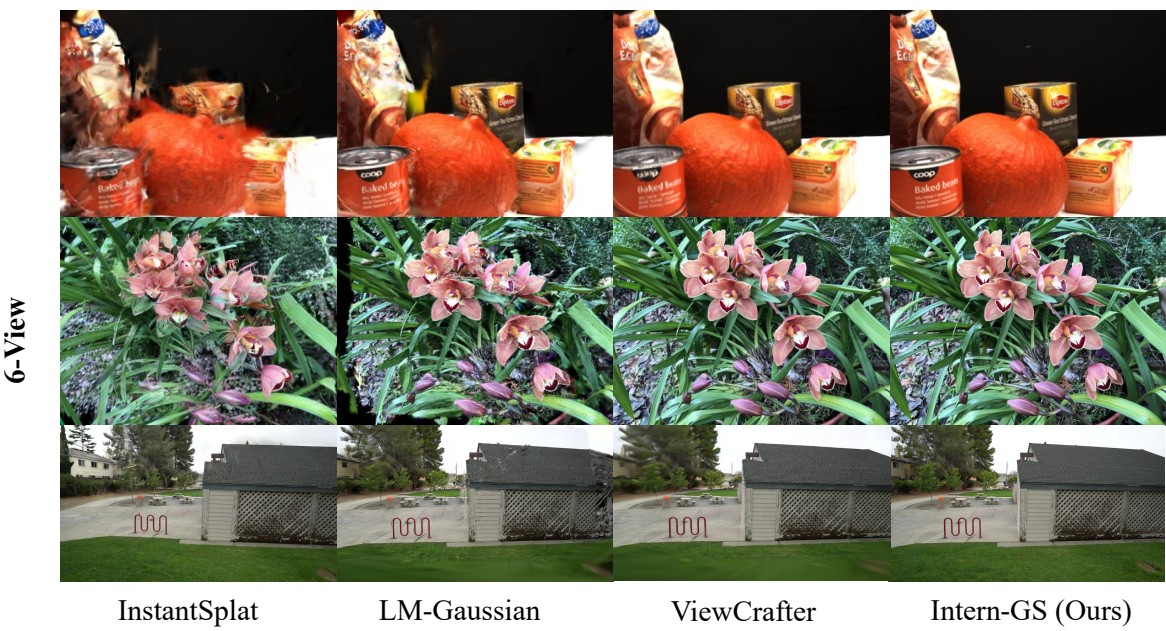

**6-View**

InstantSplat  LM-Gaussian  ViewCrafter  Intern-GS (Ours)

Figure 7: The comparison results between Instantsplat, LM-Gaussian, ViewCrafter and our method in 6-view input.

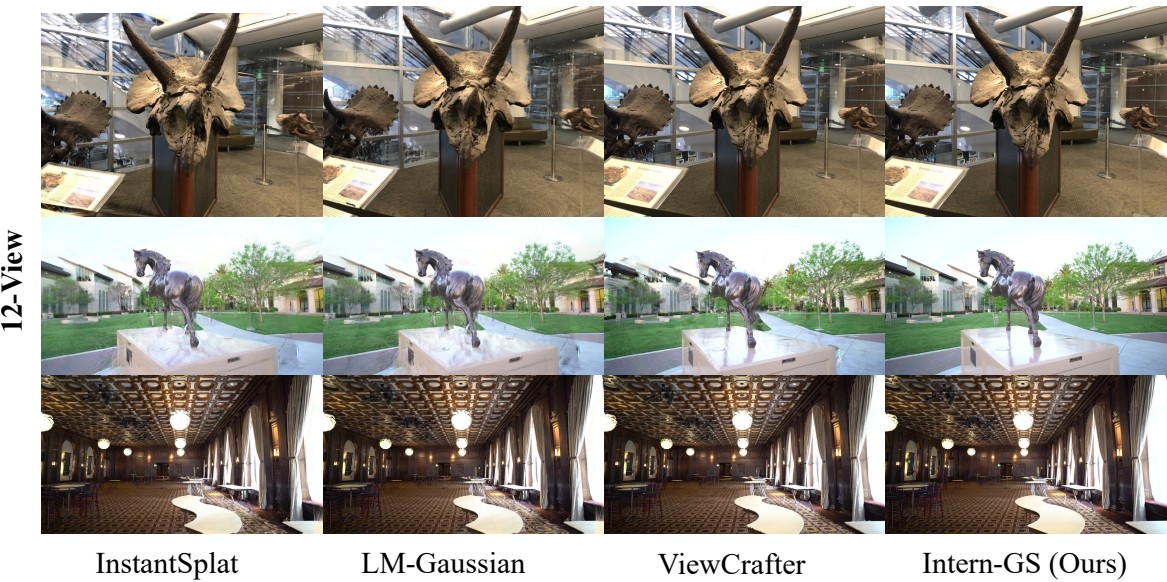

**12-View**

InstantSplat       LM-Gaussian       ViewCrafter       Intern-GS (Ours)

Figure 8: The comparison results between Instantsplat, LM-Gaussian, ViewCrafter and our method in 12-view input.

Table 7: Comparison of training time (in seconds or minutes) on the Tanks and Temples dataset under 3-view, 6-view, and 12-view settings.

| Method | 3-view | 6-view | 12-view |
|---|---|---|---|
| 3DGS Kerbl et al. (2023) | 3min47s | 7min29s | 10min36s |
| FSGS Zhu et al. (2024) | 4min21s | 8min19s | 12min30s |
| DNGaussian Li et al. (2024) | 3min56s | 7min53s | 11min49s |
| SparseGS Zhu et al. (2024) | 5min43s | 8min14s | 12min30s |
| InstantSplat Fan et al. (2024) | **10.4s** | **15.4s** | **34.1s** |
| LM-Gaussian Yu et al. (2024a) | 3min21s | 7min10s | 10min02s |
| ViewCrafter  Yu et al. (2024b) | 1min23s | 3min19s | 6min43s |
| Intern-GS (Ours) | 22s | 34s | 1min04s |

12-view setting. As the number of views increases, the influence of ground truth training view constraints becomes more significant. This may explain why multiple methods tend to produce similar performance across various metrics as the number of input views grows.

### C.3 Efficiency comparison

As shown in table 7, We compared the training time of our method with 3DGS-based approaches on the Tanks and Temples dataset under 3-view, 6-view, and 12-view settings. The results show that our method is more efficient than LM-Gaussian, and ViewCrafter. Both our method and LM-Gaussian, which used diffusion as prior do not perform better in efficiency than Instantsplat which do not use diffusion prior. But our method cost less training time than LM-Gaussian and ViewCrafter, and this advantage stems from our redundancy-free initialization strategy, which significantly reduces the burden of redundant Gaussians and leads to improved training efficiency.

Table 8: Ablation study of FSGS on Tanks and Temples with three training views, analyzing the individual contributions of the proposed two modules: Multi-View Stereo Guided Dense and non-redundant Initialization and Complete regularization method.

| Method | PSNR ↑ | LPIPS ↓ | SSIM ↑ | Training Time |
|---|---|---|---|---|
| FSGS | 22.31 | 0.197 | 0.693 | 4min21s |
| w/ DUST3R | 21.41 | 0.224 | 0.674 | 42s |
| w/ Our Init. | 22.45 | 0.196 | 0.719 | 24s |
| w/ Our Regu. | 22.38 | 0.197 | 0.708 | 4min55s |

Table 9: Ablation study of Instantsplat on Tanks and Temples with three training views, analyzing the individual contributions of the proposed two modules: Multi-View Stereo Guided Dense and non-redundant Initialization and Complete regularization method.

| Method | PSNR ↑ | LPIPS ↓ | SSIM ↑ | Training Time |
|---|---|---|---|---|
| Instantsplat | 22.20 | 0.199 | 0.743 | 10.4s |
| w/ Our Init. | 22.49 | 0.194 | 0.759 | 8.2s |
| w/ Our Regu. | 22.31 | 0.197 | 0.757 | 59s |

## C.4 More Ablation Study

As shown in the results of Table 8, we conducted additional ablation studies on the Tanks and Temples dataset based on the FSGS baseline. The first row presents the original FSGS results. The second row replaces COLMAP with DUST3R for initialization in FSGS. The third row adopts our redundancy-free initialization method. The fourth row applies our regularization strategy in place of the original regularization constraints in FSGS.

From the results, we observe that both DUST3R and our initialization method significantly reduce training time compared to the original setup. However, using DUST3R for initialization actually degrades the performance of FSGS on the Tanks and Temples dataset. This may be due to the high overlap between reference views in this dataset, which causes DUST3R to generate a large number of redundant points during initialization, leading to a decline in rendering quality. In contrast, both our initialization and regularization strategies bring consistent improvements to the baseline performance metrics.

We also conducted ablation studies on the InstantSplat baseline in Table 9. The first row shows the original InstantSplat results, the second row replaces its initialization with our redundancy-free initialization method, and the third row applies our regularization strategy. The results show that while the performance metrics are relatively similar across the three variants, both our initialization and regularization methods lead to modest but consistent improvements.

## D Limitations and Future Work

Although our method has achieved impressive results in novel view synthesis setting under sparse views, it still encounters limitations in certain scenarios. It is important not only to interpolate and generate new views within the visual region but also to expand the scene outward. Generating regions outside the current scope that are color and geometry consistent also significantly affects the rendering quality of the model. Our model struggles to ensure color and geometric consistency in these outward-expanded scenes, which may be due to the diffusion model's limited refinement capability when dealing with severely missing new views. Therefore, combining the ideas of scene reconstruction and scene generation is a future research direction for us. How to use diffusion models to effectively complete internal gaps and expand scenes outward is a question worth exploring. Our preliminary idea is to add extra geometric and color consistency constraints to the diffusion model, enabling the model to have nearly identical geometry and color in outward extrapolation as in inward interpolation.

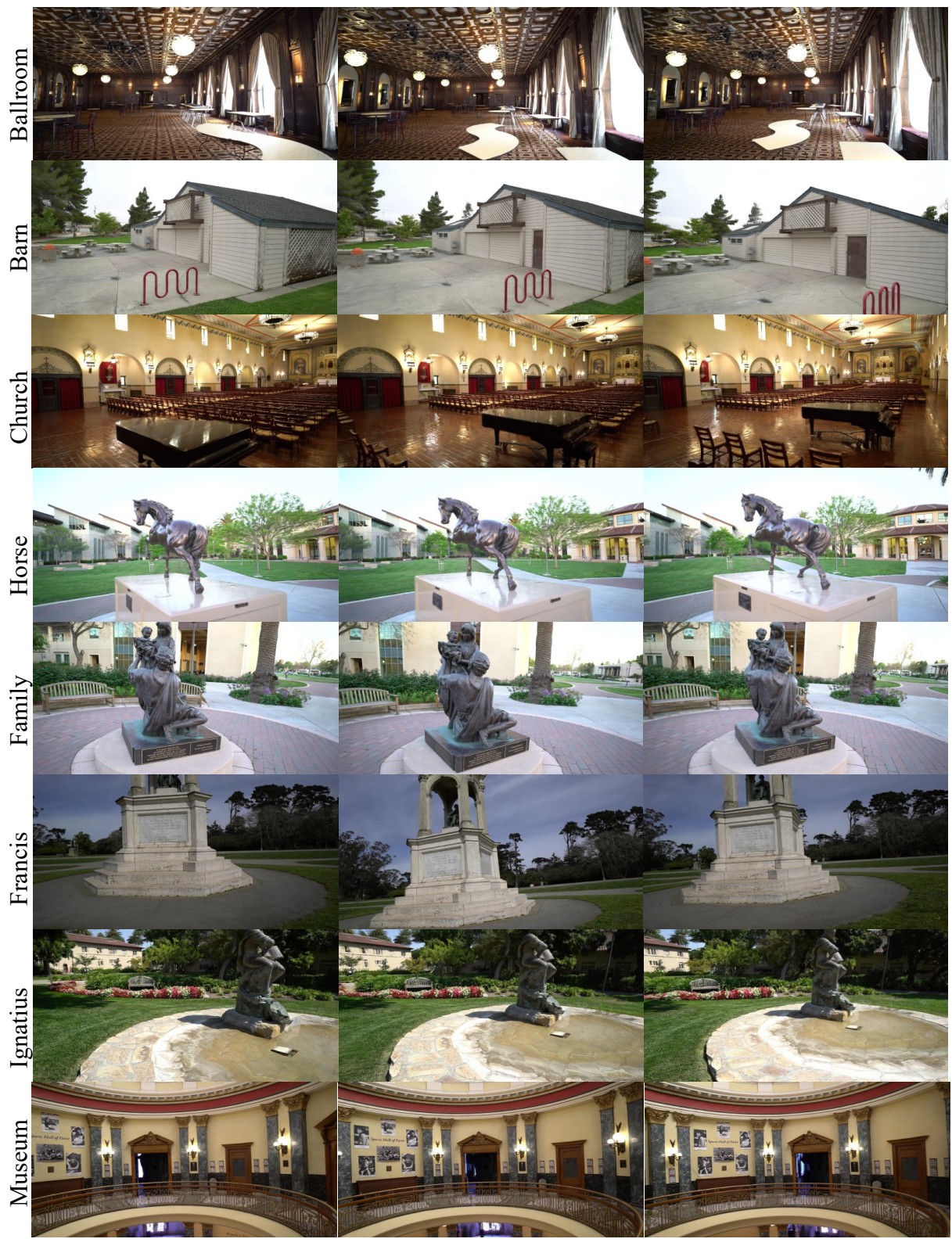

Figure 9: The qualitative results of Intern-GS on Tanks and Temples dataset under 3 training views.

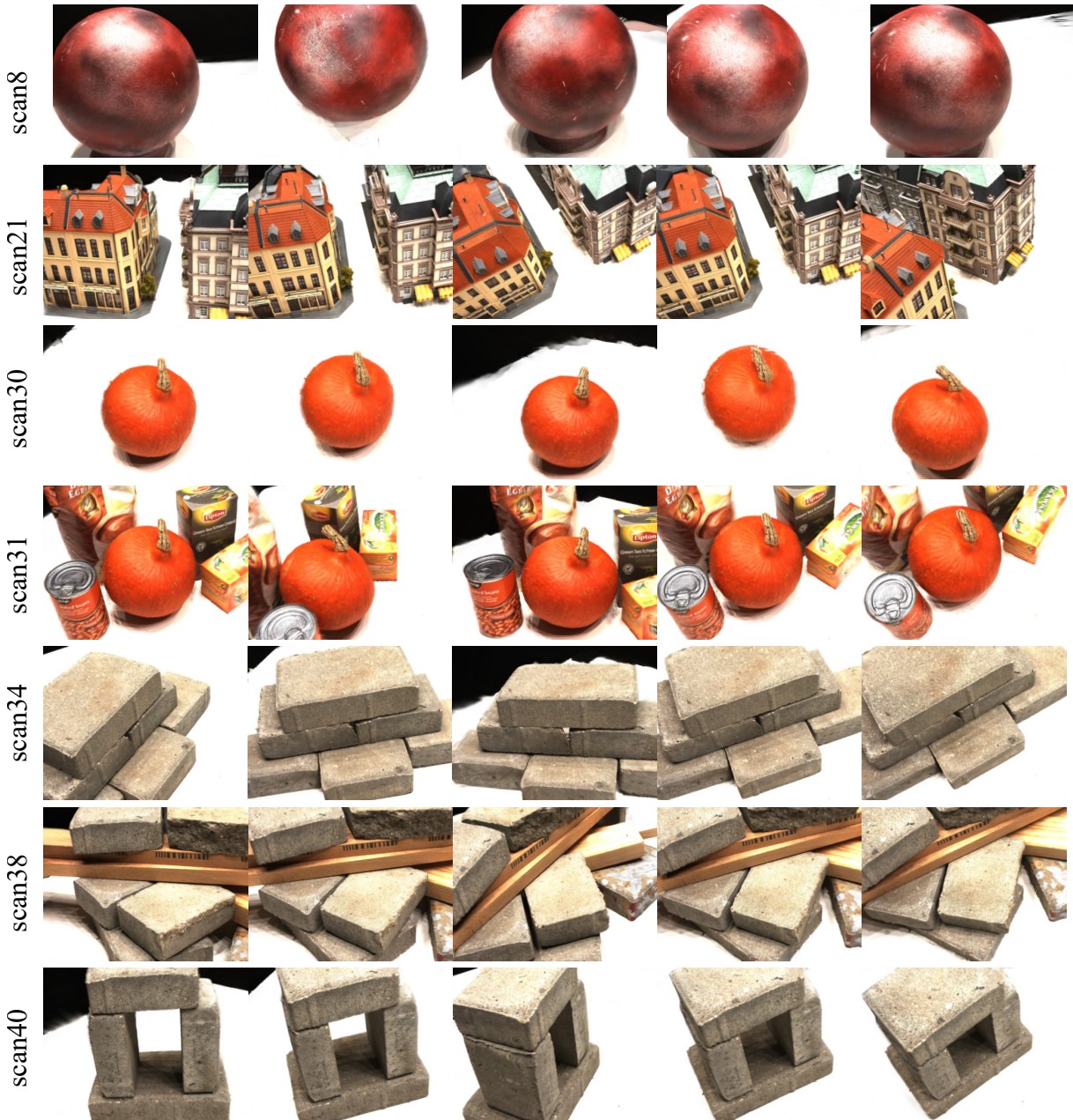

Figure 10: The qualitative results of Intern-GS on DTU dataset under 3 training views.

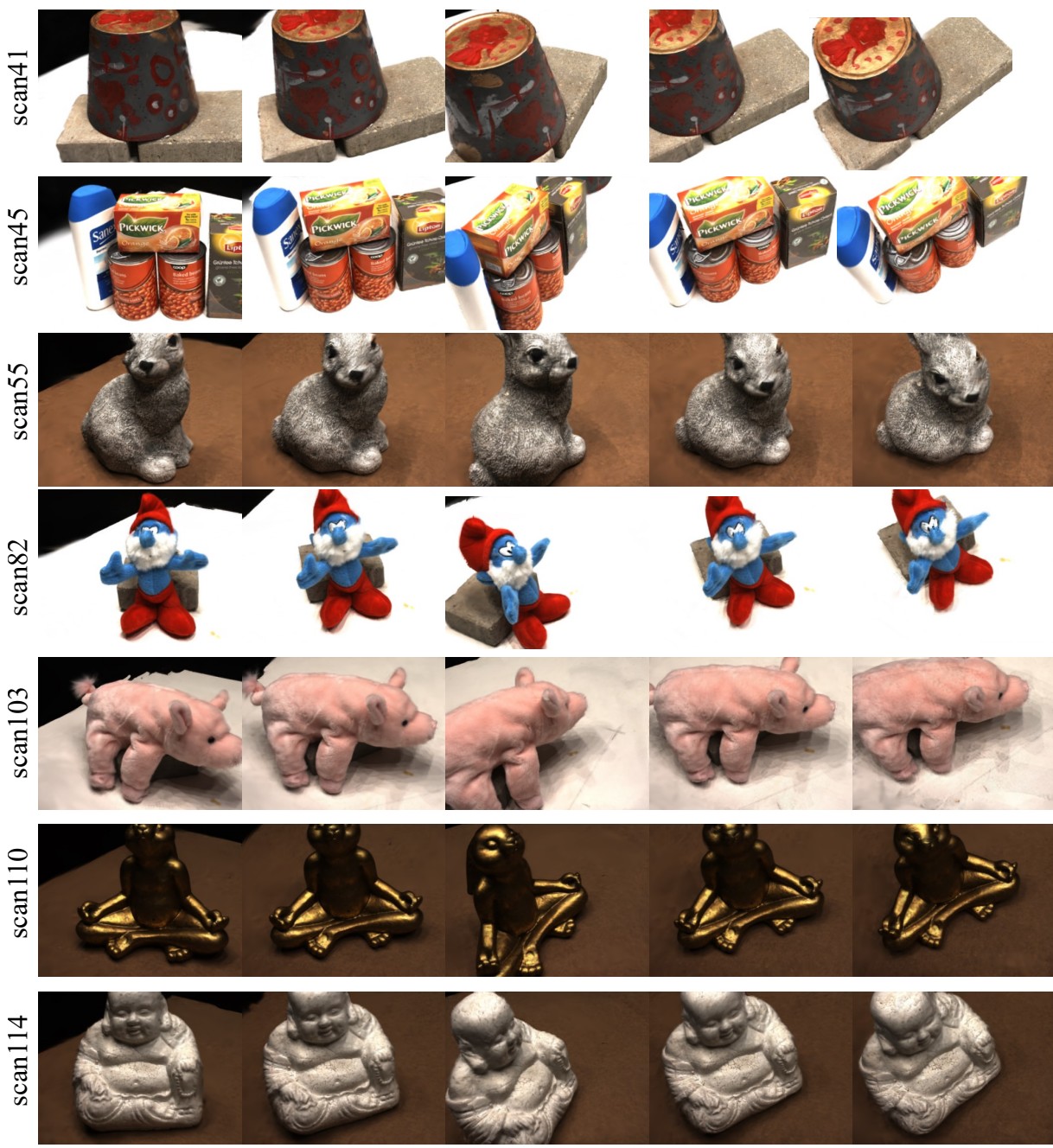

Figure 11: The qualitative results of Intern-GS on DTU dataset under 3 training views.

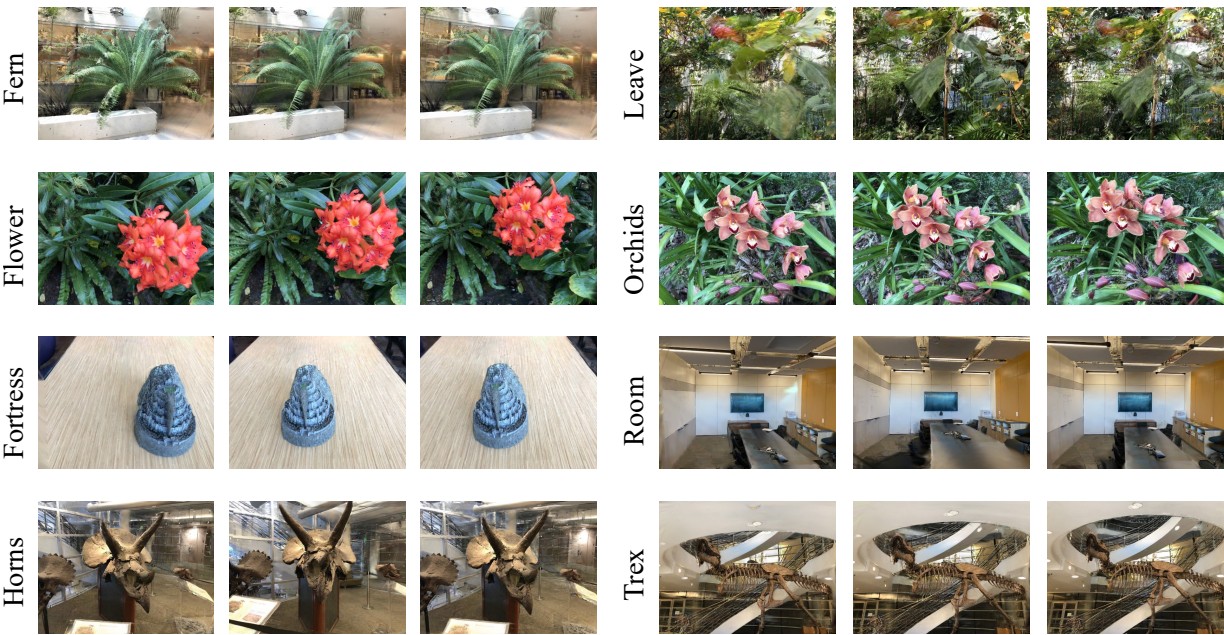

Figure 12: The qualitative results of Intern-GS on LLFF dataset under 3 training views.

