# OpenReview forum: "Intern-GS: Vision Model Guided Sparse-View 3D Gaussian Splatting"
_TMLR — Rejected by TMLR_

### Review · Reviewer_i65J · 2025-05-25

**Summary Of Contributions:**

Intern-GS replaces sparse SfM initializations with a DUSt3R-based multi-view stereo (MVS) point cloud, producing a dense, non-redundant set of 3D Gaussians that better cover texture-poor regions. It also introduces a masking mechanism to downsample overlapping points. Combines depth priors from DUSt3R and a monocular estimator with diffusion-based appearance refinement to enforce geometric and photometric consistency on pseudo views to improve the uncertain region. Extensive experiments were conducted on LLFF, DTU, and Tanks & Temples benchmarks under only three input views.

**Audience:**

No

**Claims And Evidence:**

No

**Requested Changes:**

- Could authors provide more validation in terms of if Dus3tr is the best prior for the pipeline?
- Could the authors justify if the used metrics are meaningful?

**Strengths And Weaknesses:**

## Strength

- Intuitive Use of Foundation Models

  Leveraging DUSt3R and diffusion priors directly for both initialization and optimization bridges the gap between data-driven and classical geometry approaches.

- Comprehensive Experimental Validation

  Evaluation across forward-facing and non-frontal scenes (LLFF, DTU, Tanks & Temples) with both quantitative metrics and qualitative examples convincingly supports the claimed gains.


## Weakness

- While the paper’s experiments are extensive, it largely overlooked the thread of related work in feedforward reconstruction approaches [A][B][C]. In theory, these approaches view the reconstruction problem similar to DUS3TR, and can be amended with diffusion prior as well. In theory, such an initialization approach is more favorable than DUS3TR due to the Gaussian representation. Could the authors discuss and experiment with these priors?

  [A] Jin, H., Jiang, H., Tan, H., Zhang, K., Bi, S., Zhang, T., ... & Xu, Z. (2024). Lvsm: A large view synthesis model with minimal 3d inductive bias. arXiv preprint arXiv:2410.17242.

  [B] Zhang, K., Bi, S., Tan, H., Xiangli, Y., Zhao, N., Sunkavalli, K., & Xu, Z. (2024, September). Gs-lrm: Large reconstruction model for 3d gaussian splatting. In European Conference on Computer Vision (pp. 1-19). Cham: Springer Nature Switzerland.

  [C] Chen, Y., Wang, J., Yang, Z., Manivasagam, S., & Urtasun, R. (2024, September). G3r: Gradient guided generalizable reconstruction. In European Conference on Computer Vision (pp. 305-323). Cham: Springer Nature Switzerland.

- Moreover, one of the biggest concern is the evaluation. The use of reconstruction metrics like PSNR in novel views are inherently ill-posed due to the multi-modal distribution. It is impossible to evaluate the quality with a fixed set of views when diffusion sampling differs from the ground truth. Could authors comment on why shall we care about such an evaluation approach?
- Lastly, the paper seems to be a system paper that combines prior work together, without any algorithm/machine learning innovation. Thus, it may not be suitable for TMLR. Could authors comment on this?

---

> ### Author Response · Authors · 2025-06-12
> **Response to Reviewer i65J**
>
> We sincerely thank the reviewer for recognizing the comprehensiveness of our experiments. In response to the specific concerns, we offer the following clarifications:
>
> ---
>
> ### **1. Overlooking related work in feedforward approaches (LVSM, GS-LRM, G3R)**
>
> First, we would like to clarify our task setup: Input: Multi-view RGB images; Output: Novel-view images and 3DGS representations. Although LVSM, GS-LRM, and G3R are feedforward models similar to ours, they require significantly more condition as input:
>
> - LVSM: Requires explicit pose information and only generates novel views, not 3DGS.
> - GS-LRM: Also requires pose as input.
> - G3R: Needs both poses and a geometry scaffold.
>
> In contrast, our method requires only multi-view images and no pose or geometry prior. Therefore, these methods cannot replace DUST3R in our pipeline, which highlights the greater flexibility of our approach.
>
> To further validate this, we compared the reconstruction quality of our method with these models: For LVSM and GS-LRM, we used COLMAP to precompute poses from 3 views; for G3R, we used DUST3R to generate the geometry scaffold.
>
> | Method  | Input                   | PSNR ↑ (LLFF) | SSIM ↑ (LLFF) | LPIPS ↓ (LLFF) | PSNR ↑ (DTU) | SSIM ↑ (DTU) | LPIPS ↓ (DTU) |
> |---------|-------------------------|---------------|----------------|----------------|---------------|----------------|----------------|
> | LVSM    | Multi-view + Pose       | 17.41          | 0.542          | 0.312          | 16.97         | 0.749        | 0.264          |
> | GS-LRM  | Multi-view + Pose       | 18.74          | 0.596         | 0.281          | 16.44          | 0.742          | 0.301          |
> | G3R     | Multi-view + Geo | 19.77          | 0.621         | 0.255          | 18.70          | 0.794          | 0.214          |
> | Ours    | Multi-view only         | **20.49**      | **0.693**      | **0.212**      | **20.34**      | **0.851**      | **0.163**      |
>
>
>
> On both LLFF and DTU datasets, under unified novel-view settings, the results showed: LVSM and GS-LRM struggle due to poor sparse-view COLMAP poses. G3R, though improved by its geometry scaffold prior, still lags behind our method. This demonstrates that, although feed-forward models generally have better generalization and efficiency, our scene-specific optimization method achieves superior reconstruction accuracy on specific scene.
>
> ---
>
> ### **2. Use of PSNR for evaluating novel-view synthesis**
>
> We thank the reviewer for raising this important concern regarding the use of reconstruction metrics such as PSNR for evaluating novel view synthesis. We respond from two main perspectives:
>
> First, when using diffusion models, there is a strong underlying assumption that the model has already encountered similar scenes — in other words, the data used in our experiments are in-domain. As a result, the randomness of diffusion sampling is heavily constrained and unlikely to produce content unrelated to the scene.
>
> Second, we apply diffusion-based color supervision only to novel views generated under extremely sparse-view conditions, where the reconstruction quality is notably poor. For training views that are already supervised by ground-truth images, we impose no additional constraints.
>
> Therefore, the task fundamentally remains a reconstruction problem, and the use of metrics such as PSNR is appropriate. There is no need to consider diversity in generation in this context.
>
> ---
>
> ### **3. Concern about lack of algorithm or ML innovation**
>
> We sincerely apologize for the confusion and would like to take this opportunity to restate our contributions.
>
> First, we do not directly use DUST3R for initialization. Instead, we design a new redundancy removal strategy built upon it, which we name RF (Redundancy-Free). This module leverages per-frame confidence information to eliminate Gaussians corresponding to well-initialized regions from previous frames. The original DUST3R tends to produce redundant initializations with many overlapping Gaussian ellipsoids, which significantly impacts optimization efficiency. We have revised the manuscript and added a new section in the supplementary material to quantitatively demonstrate the effectiveness of this module (Appendix C.2, C.3, C.4) .
>
> In addition, we introduce novel color and depth refinement strategies specifically designed for pseudo-views, which distinguish our approach from other 3DGS methods that rely on pretrained models. For example, InstantSplat does not incorporate any refinement mechanism for pseudo-views, while LM-Gaussian applies a diffusion-driven regularization only to the color. In contrast, our method explicitly targets both factors that affect 3DGS quality: color and geometry, by introducing corresponding regularization strategies for each.
>
> ---
>
> We hope these clarifications help underline the technical novelty and practical value of our work. We would be happy to clarify any additional questions you may have, please feel free to reach out.

---

### Review · Reviewer_7BKo · 2025-05-27

**Summary Of Contributions:**

This paper introduces Intern-GS, a hybrid methodology designed to enhance novel view synthesis from sparse inputs by leveraging multiple pre-trained vision foundation models. The approach initiates Gaussian Splatting with a dense point cloud derived from DUSt3R. To bolster supervision during optimization, Intern-GS incorporates two key strategies: it generates additional novel views using a pre-trained multi-view diffusion model and enforces geometric consistency through depth supervision from a pre-trained depth estimation model. Comprehensive evaluations on established benchmarks demonstrate that Intern-GS achieves superior performance in synthesizing novel views from sparse view images.

**Audience:**

No

**Claims And Evidence:**

No

**Requested Changes:**

See weaknesses above. In its current form, and without the critical revisions detailed above (particularly concerning novelty and experimental validation), the manuscript does not meet the threshold for acceptance.

**Strengths And Weaknesses:**

**Strengths**
- The paper is well-written, with a clear structure that makes the methodology and results easy to comprehend. Sufficient details are provided to understand the proposed approach.
- The Intern-GS method demonstrates state-of-the-art (SOTA) performance on several widely recognized benchmarks for novel view synthesis, indicating its effectiveness.
- The paper addresses the task of novel view synthesis from sparse inputs, a challenging yet highly relevant problem with significant practical implications for real-world applications where data capture is often limited

**Weaknesses**
- While the paper presents an integrated system, the primary concern lies in the **novelty of its constituent parts**. The core techniques—utilizing DUSt3R for Gaussian Splatting initialization and employing diffusion models for view augmentation—have been explored in prior works. The paper could benefit from a clearer articulation of how this specific combination yields non-obvious synergistic benefits beyond an incremental improvement.
- The "Method" section (Section 3) dedicates considerable space to describing pre-existing techniques like DUSt3R (Sec. 3.2) and diffusion models (Sec. 3.4). It is recommended to relocate this background information to a "Preliminaries" or "Related Work" section. More importantly, the paper needs to **explicitly differentiate** its novel contributions from established methods to clearly highlight its unique advancements.
- The claimed contributions (DUSt3R initialization, diffusion and depth model supervision) appear to be applications of existing tools rather than new concepts. Given that many similar approaches have been previously investigated, the inventive step of this work is unclear. The paper should better justify why this particular integration is significant and offers new insights to the field.
- While Intern-GS achieves SOTA results, the performance margins over some baselines (e.g., FSGS, InstantSplat) appear marginal in certain tables (Table 1 and Table 2). This raises questions about the source of the performance gains. To provide a more convincing evaluation, the authors should include ablation studies or comparisons where baselines are enhanced with similar components (e.g., FSGS + DUSt3R initialization, InstantSplat + depth supervision). This would help isolate the specific benefits derived from the Intern-GS framework itself.
- The experiments are currently limited to a 3-view setting. To demonstrate robustness and broader applicability, the evaluation should be extended to include other sparse-view scenarios, such as 6-view or 12-view settings.
- The paper lacks a discussion on the training time and computational resources required by Intern-GS. Including a comparison with other methods in terms of efficiency would provide a more complete picture of its practical viability.
- There are minor instances where the first-person singular ("I") is used (e.g., in Section 4). These should be revised to the conventional first-person plural ("we") for consistency in academic writing.

---

> ### Author Response · Authors · 2025-06-12
> **Response to Reviewer 7BKo (Part1)**
>
> We sincerely thank the reviewer for their recognition of the structure and writing quality of our manuscript. We also greatly appreciate the constructive suggestions provided to help us improve the paper. Below, we address each of your concerns in detail. Please find our point-by-point responses below.
>
> ---
>
> ## 1. The primary concern lies in the novelty of its constituent parts
>
> We acknowledge that our original manuscript did not sufficiently highlight the novelty of our approach. In the revised version, we made substantial updates to the following sections: Abstract, Contributions, Related Work, Method.
>
> We would also like to further emphasize our contributions and highlight the differences between our method and recent approaches that are likewise based on visual foundation models (Instantsplat, LM-Gaussian). Our key novelties include:
>
> - We do not directly use DUST3R. Instead, we propose a **confidence-based redundancy removal module (RF)**, which eliminates redundant 3D Gaussians and significantly improves rendering efficiency. Naively using DUST3R introduces heavy Gaussian redundancy, which severely impacts optimization and rendering.
> - We design a **hybrid regularization strategy** that jointly constrains color and geometry — two critical aspects of Gaussian quality in both training view and pseudo view. In contrast:
>   - InstantSplat does not consider pseudo-view color and depth regularization.
>   - LM-Gaussian does not consider pseudo-view depth regularization.
>
> Thus, even though the pipelines may appear similar, our method integrates more complete and principled constraints.
>
> ---
>
> ## 2. The "Method" section (Section 3) dedicates considerable space to describing pre-existing techniques
>
> We have rewritten the DUST3R portion and remove the DUST3R part to related work to focus on our innovations, especially the **redundancy removal module**. We retain the brief introduction of diffusion-based priors in the color optimization section to highlight how we combine multiple types of regularization into a unified framework — a key difference from previous works that typically lack certain constraints.
>
> ---
>
> ## 3. To provide a more convincing evaluation, the authors should include ablation studies or comparisons where baselines are enhanced with similar components.
>
> We appreciate this suggestion. We conducted additional ablation studies by injecting our redundancy-free initialization into FSGS and our hybrid regularization into InstantSplat.
>
> | Method         | PSNR ↑ | LPIPS ↓ | SSIM ↑ | Training Time |
> |----------------|--------|---------|--------|----------------|
> | FSGS           | 22.31  | 0.197   | 0.693  | 4min21s        |
> | w/ DUST3R      | 21.41  | 0.224   | 0.674  | 42s            |
> | w/ Our Init.   | 22.45  | 0.196   | 0.719  | 24s            |
> | w/ Our Regu.   | 22.38  | 0.197   | 0.708  | 4min55s        |
>
> As shown in the table above, we conducted additional ablation studies on the Tanks and Temples dataset based on the FSGS baseline. The first row presents the original FSGS results. The second row replaces COLMAP with DUST3R for initialization in FSGS. The third row adopts our redundancy-free initialization method. The fourth row applies our regularization strategy in place of the original regularization constraints in FSGS. From the results, we observe that both DUST3R and our initialization method significantly reduce training time compared to the original setup. However, using DUST3R for initialization actually degrades the performance of FSGS on the Tanks and Temples dataset. This may be due to the high overlap between reference views in this dataset, which causes DUST3R to generate a large number of redundant points during initialization, leading to a decline in rendering quality. In contrast, both our initialization and regularization strategies bring consistent improvements to the baseline performance metrics.
>
> | Method         | PSNR ↑ | LPIPS ↓ | SSIM ↑ | Training Time |
> |----------------|--------|---------|--------|----------------|
> | InstantSplat   | 22.20  | 0.199   | 0.743  | 10.4s          |
> | w/ Our Init.   | 22.49  | 0.194   | 0.759  | 8.2s           |
> | w/ Our Regu.   | 22.31  | 0.197   | 0.757  | 59s            |
>
> We also conducted ablation studies on the InstantSplat baseline in the table above. The first row shows the original InstantSplat results, the second row replaces its initialization with our redundancy-free initialization method, and the third row applies our regularization strategy. The results show that while the performance metrics are relatively similar across the three variants, both our initialization and regularization methods lead to modest but consistent improvements.
>
> We have updated this part in Appendix C.4 of the revised manuscript.

---

> ### Author Response · Authors · 2025-06-12
> **Response to Reviewer 7BKo (Part2)**
>
> ## 4. The experiments are currently limited to a 3-view setting
>
> We have conducted additional experiments under the 6-view and 12-view settings on the Tanks and Temples dataset, following InstantSplat’s experimental protocol. The results are shown below:
>
>
> | Method             | PSNR ↑ (6-view) | PSNR ↑ (12-view) | SSIM ↑ (6-view) | SSIM ↑ (12-view) | LPIPS ↓ (6-view) | LPIPS ↓ (12-view) |
> |--------------------|-----------------|------------------|------------------|------------------|------------------|-------------------|
> | 3DGS               | 17.78           | 18.92            | 0.593            | 0.621            | 0.349            | 0.317             |
> | FSGS               | 24.41           | 27.87            | 0.768            | 0.861            | 0.153            | 0.129             |
> | DNGaussian         | 23.34           | 25.79            | 0.783            | 0.814            | 0.258            | 0.219             |
> | SparseGS           | 23.56           | 26.04            | 0.792            | 0.831            | 0.229            | 0.187             |
> | InstantSplat       | 25.45           | 28.19            | 0.8453           | 0.8785           | 0.1173           | **0.1068**        |
> | LM-Gaussian        | 25.68           | 28.34            | 0.851            | 0.881            | 0.117            | 0.113             |
> | ViewCrafter        | 25.62           | 28.23            | 0.832            | 0.877            | 0.121            | 0.118             |
> | **Intern-GS (Ours)** | **26.41**     | **28.59**        | **0.863**        | **0.882**        | **0.112**        | 0.111             |
>
> The results show that our method still demonstrates a significant advantage under the 6-view setting. However, as the number of input views increases, the influence of ground truth training view constraints becomes more pronounced. This may explain why different methods tend to yield similar performance across various metrics when the number of views grows.
>
> We have updated this part in Appendix C.2 of the revised manuscript.
>
> ---
>
> ## 5. The paper lacks a discussion on the training time and computational resources required by Intern-GS
>
> We have compared the training efficiency of Intern-GS with prior 3DGS-based methods:
>
> | Method          | 3-view   | 6-view   | 12-view   |
> |------------------|----------|----------|-----------|
> | 3DGS             | 3min47s  | 7min29s  | 10min36s  |
> | FSGS             | 4min21s  | 8min19s  | 12min30s  |
> | DNGaussian       | 3min56s  | 7min53s  | 11min49s  |
> | SparseGS         | 5min43s  | 8min14s  | 12min30s  |
> | InstantSplat     | **10.4s**| **15.4s**| **34.1s** |
> | LM-Gaussian      | 3min21s  | 7min10s  | 10min02s  |
> | ViewCrafter      | 1min23s  | 3min19s  | 6min43s   |
> | Intern-GS (Ours) | 22s      | 34s      | 1min04s   |
>
> - Compared to COLMAP-based pipelines, our approach significantly reduces time due to feed-forward geometry prediction.
> - Compared to InstantSplat, our method incurs additional overhead due to diffusion regularization — however, this is mitigated by our redundancy-free initialization.
> - Compared to LM-Gaussian, which also used diffusion as prior fairly, our method is faster overall, as we start from a non-redundant initialization.
>
> We have updated this part in Appendix C.3 of the revised manuscript.
>
> ---
>
> ## 6. There are minor instances where the first-person singular ("I") is used (e.g., in Section 4)
>
> We have carefully reviewed the manuscript and corrected all occurrences of “I” to “we” to maintain a consistent and professional tone throughout the paper.
>
> ---
>
> We hope these clarifications help underline the technical novelty and practical value of our work. Should you have any further questions or suggestions, we would be happy to respond. Please feel free to reach out.

---

### Review · Reviewer_8crj · 2025-05-29

**Summary Of Contributions:**

The authors introduce Intern-GS, a method for sparse-view novel view synthesis based on 3D Gaussian Splatting. This is achieved through a dense Gaussian initialization strategy (by DUSt3R) guided by multi-view stereo priors from vision foundation models, which notably improves performance in texture-poor regions. Furthermore, the work presents a hybrid regularization mechanism that utilizes pre-trained diffusion models for refining appearance and deep depth estimation models for imposing depth constraints, leading to more consistent and accurate reconstructions from unobserved viewpoints. Extensive experiments confirm that Intern-GS achieves state-of-the-art results on several challenging datasets

**Audience:**

Yes

**Broader Impact Concerns:**

While advanced novel view synthesis offers benefits, its capacity to create high-fidelity visual content from sparse data raises concerns about potential misuse for misinformation. Additionally, the dependence on large vision foundation models introduces challenges related to significant computational and environmental costs, potentially limiting broader accessibility.

**Claims And Evidence:**

Yes

**Requested Changes:**

While the paper demonstrates strong results, a more explicit differentiation from recent works could strengthen its positioning. For instance, a clearer articulation of Intern-GS's advantages over methods like InstantSplat/LM-Gaussian, particularly concerning aspects beyond the reported metrics, such as initialization robustness or optimization efficiency, would be beneficial. And the techniques using diffusion model against ViewCrafter.

The authors should also consider a more direct discussion on why certain very recent visual foundation model-based 3DGS methods were not included in visual comparisons.

Report the optimization duration and rendering speed would be good for understanding the efficiency.

**Strengths And Weaknesses:**

The paper’s strongest contributions and aspects include:

- A dense and non-redundant Gaussian initialization strategy using the DUSt3R model.
- An innovative hybrid regularization mechanism which combines pre-trained diffusion models for appearance refinement with deep depth estimation models for geometric constraints, effectively enriching the detail and consistency of unobserved viewpoints.

However, the work also presents some significant weaknesses:

The manuscript could better highlight its specific novelty, as individual techniques seems draw from existing research; clearly distinguishing the unique advantages of its DUSt3R-guided initialization, diffusion-based enhancement, and hybrid regularization framework  over other recent foundation model-based 3DGS methods is crucial. Additionally, visual comparisons would be more convincing if they consistently included strong numerical baselines (which is benchmarked in tables but not always in key visual figures ), to visually affirm the entire pipeline's superiority beyond component-level ablation studies.

---

> ### Author Response · Authors · 2025-06-12
> **Response to Reviewer 8crj (Part1)**
>
> We sincerely thank the reviewers for their constructive feedback. In response to the comments and concerns raised, we have revised the manuscript accordingly and provide detailed clarifications below:
>
> ### 1. A more explicit differentiation from recent works could strengthen its positioning.
>
> While our framework shares a similar overall structure with recent 3DGS methods based on vision foundation models, there are critical differences in module design.
>
> - First, we do not simply adopt DUST3R for initialization. A naive application of DUST3R would produce highly redundant point clouds, as the input images often contain overlapping content, and DUST3R operates as a pixel-to-pixel prediction model. This overlap leads to the same regions being repeatedly reconstructed, resulting in a large number of redundant points. To address this, we propose the **Redundancy-Free (RF)** strategy, which utilizes confidence maps to remove such redundant regions. This approach is fundamentally different from methods like InstantSplat, LM-Gaussian, and ViewCrafter.
>
> - In the regularization phase, InstantSplat only considers MVS priors, LM-Gaussian and ViewCrafter adopt diffusion-based visual priors but ignore depth constraints in pseudo views. Our method introduces a more comprehensive regularization scheme, enforcing both color and geometric consistency across training and pseudo views.
>
> Also, we have revised the Related Work section and Contribution part accordingly to include these methods and highlight our methodological advantages.
>
> ---
>
> ### 2. The authors should also consider a more direct discussion on why certain very recent visual foundation model-based 3DGS methods were not included in visual comparisons.
>
> We appreciate this insightful suggestion. Based on our investigation, we included InstantSplat, LM-Gaussian, and ViewCrafter as new comparison baselines. In the supplementary material, we provide both quantitative and visual comparisons with these methods. We have:
>
> - Added visual comparisons in the supplementary material featuring InstantSplat, LM-Gaussian, ViewCrafter and our method.
> - Updated Table 1 and Table 2 in the main paper to include numerical results for LM-Gaussian and ViewCrafter.
> - We have additionally conducted experiments under the 6-view and 12-view settings on three datasets. The results of these experiments have been updated in Appendix C.2 of the manuscript.
>
> | Method        | PSNR ↑ (6-view) | PSNR ↑ (12-view) | SSIM ↑ (6-view) | SSIM ↑ (12-view) | LPIPS ↓ (6-view) | LPIPS ↓ (12-view) |
> |---------------|----------------|------------------|------------------|------------------|------------------|-------------------|
> | 3DGS          | 17.78          | 18.92            | 0.593            | 0.621            | 0.349            | 0.317             |
> | FSGS          | 24.41          | 27.87            | 0.768            | 0.861            | 0.153            | 0.129             |
> | DNGaussian    | 23.34          | 25.79            | 0.783            | 0.814            | 0.258            | 0.219             |
> | SparseGS      | 23.56          | 26.04            | 0.792            | 0.831            | 0.229            | 0.187             |
> | InstantSplat  | 25.45          | 28.19            | 0.8453           | 0.8785           | 0.1173           | **0.1068**        |
> | LM-Gaussian   | 25.68          | 28.34            | 0.851            | 0.881            | 0.117            | 0.113             |
> | ViewCrafter   | 25.62          | 28.23            | 0.832            | 0.877            | 0.121            | 0.118             |
> | **Intern-GS (Ours)** | **26.41** | **28.59**        | **0.863**        | **0.882**        | **0.112**        | 0.111             |
>
> Here are the results on Tanks and Temples datasets. Our method continues to demonstrate superior performance compared to InstantSplat, LM-Gaussian, and ViewCrafter under the 6-view and 12-view settings. Nevertheless, the performance margin decreases as the number of views increases. A possible explanation is that the increased number of GT training views introduces stronger optimization bias, which in turn reduces the performance gap.

---

> ### Author Response · Authors · 2025-06-12
> **Response to Reviewer 8crj (Part2)**
>
> ---
>
> ### 3. Reporting the optimization duration and rendering speed would be good for understanding the efficiency.
> Thanks for the suggestion. We performed additional experiments on the Tanks and Temples dataset under the 3-view setting, comparing training time across the following methods: 3DGS, FSGS, DNGaussian, SparseGS, InstantSplat, LM-Gaussian, and ViewCrafter. Here is the results on Tanks and Temples dataset,
>
> | Method         | 3-view   | 6-view   | 12-view   |
> |----------------|----------|----------|-----------|
> | 3DGS           | 3min47s  | 7min29s  | 10min36s  |
> | FSGS           | 4min21s  | 8min19s  | 12min30s  |
> | DNGaussian     | 3min56s  | 7min53s  | 11min49s  |
> | SparseGS       | 5min43s  | 8min14s  | 12min30s  |
> | InstantSplat   | **10.4s** | **15.4s** | **34.1s** |
> | LM-Gaussian    | 3min21s  | 7min10s  | 10min02s  |
> | ViewCrafter    | 1min23s  | 3min19s  | 6min43s   |
> | **Intern-GS (Ours)** | 22s      | 34s      | 1min04s   |
>
> Compared to methods that rely on SfM-based initialization, our approach demonstrates a clear advantage in training efficiency. While InstantSplat achieves the fastest training speed overall due to the absence of any diffusion-based refinement modules, our method still outperforms LM-Gaussian and ViewCrafter—both of which incorporate diffusion modules—by a significant margin. Specifically, on the 3-view setting, our method is 3 minutes and 1 minute faster than LM-Gaussian and ViewCrafter, respectively. On the 6-view setting, we achieve speedups of 6.5 minutes and 2.5 minutes, respectively. In the 12-view scenario, our method is 9 minutes faster than LM-Gaussian and 6.5 minutes faster than ViewCrafter. We have updated the appendix to include these experimental results, you can find in Table 7.
>
> ---
>
> We hope these clarifications help underline the technical novelty and practical value of our work. Should you have any further questions or suggestions, we would be happy to respond. Please feel free to reach out.

---

### Decision · Action_Editor_c6m3 · 2025-07-07

**Recommendation:** Reject

**Additional Comments:**

While I feel it would be unjust to eliminate possibility of a resubmission, I have concerns about whether it will be acceptable due to the problems in positioning the work after the major revision that occurred in response to reviewer comments and the major clarity / quality issues that remain, a few of which I highlighted.  Also, while the paper is technically in scope, it would be better suited for a conference like 3DV which would have a larger pool of expert reviewers in this topic.

**Audience:**

Yes

**Audience Explanation:**

However, in the current version it is difficult to pinpoint the new ideas due to lack of clarity in the text and insufficient positioning with respect to existing work.  A major revision would be needed.

**Claims And Evidence:**

No

**Claims Explanation:**

The reviewers consistently raise concerns that the work does not sufficiently establish its contributions over related works. While TMLR does not consider novelty to be a major factor in decisions, the reader should be able to easily understand what can be learned from the paper.  The original submission was heavily edited during the review process to try to make this more clear, but the reviewers are still not convinced and recommend reject.

Two main contributions are claimed:
1. Remove redundant points from Dust3r pointmaps.  While the method may differ from other implementations, there are many methods for doing depth map fusion that remove redundant and inconsistent points.  There is no evidence that the heuristic method presented is any better than them.
2. "Hybrid regularization" combining multiple losses: It's not clear in what respects this is significantly different than, e.g. MonoPatchNeRF that combines monocular depth/normal, virtual views, and photometric losses.

Although one reviewer says that the paper is easy to read, the action editor disagrees.  The paper needs better internal quality control, e.g.
* Figures 1, 2, and 3 are not referenced by the text.
* Figure 2 inexplicably compares SfM point clouds to their Dust3r based initialization and cites Ulman '79 as the SfM method, rather than the method used to generate the SfM point clouds. I say inexplicably because SfM does not attempt to generate dense point clouds, while MVS and Dust3r and other methods do.  The caption says "Our method outperforms SfM in low-texture areas", which falsely implies that the aim of SfM is to generate a point cloud (the aim is to register images).
* Fig 3, the main method figure includes a box for "Vision-Language Model Feature".  There is no language in the method, and it's not clear what this refers to.  It's not described in the text.
* Section titles in 2.1 and 2.2 misspell "Synthesis".  There are numerous other typos.
* The paper is sometimes confusing about whether it is aiming for 3D reconstruction (i.e. point cloud generation) or novel view synthesis.  E.g. Fig 1 refers to "reconstruction results", Fig 2 claims their method generates better point clouds than SfM. Clearly, though the aim and evaluation is for novel view synthesis rather than accurate 3d point clouds, as the geometric accuracy is not evaluated.

**Resubmission Of Major Revision:**

The authors may consider submitting a major revision at a later time.